# TFEB regulates sulfur amino acid and coenzyme A metabolism to support hepatic metabolic adaptation and redox homeostasis

David Matye[1,2], Sumedha Gunewardena[3], Jianglei Chen[1], Huaiwen Wang[4], Yifeng Wang[2], Mohammad Nazmul Hasan [1], Lijie Gu[1], Yung Dai Clayton[1], Yanhong Du[1], Cheng Chen[1], Jacob E. Friedman[1], Shelly C. Lu [5], Wen-Xing Ding [2] & Tiangang Li [1]✉

Fatty liver is a highly heterogenous condition driven by various pathogenic factors in addition to the severity of steatosis. Protein insufficiency has been causally linked to fatty liver with incompletely defined mechanisms. Here we report that fatty liver is a sulfur amino acid insufficient state that promotes metabolic inflexibility via limiting coenzyme A availability. We demonstrate that the nutrient-sensing transcriptional factor EB synergistically stimulates lysosome proteolysis and methionine adenosyltransferase to increase cysteine pool that drives the production of coenzyme A and glutathione, which support metabolic adaptation and antioxidant defense during increased lipid influx. Intriguingly, mice consuming an isocaloric protein-deficient Western diet exhibit selective hepatic cysteine, coenzyme A and glutathione deficiency and acylcarnitine accumulation, which are reversed by cystine supplementation without normalizing dietary protein intake. These findings support a pathogenic link of dysregulated sulfur amino acid metabolism to metabolic inflexibility that underlies both overnutrition and protein malnutrition-associated fatty liver development.

It has been increasingly recognized that non-alcoholic fatty liver disease (NAFLD) is a highly heterogenous condition and its progression is driven by various pathogenic factors in addition to the severity of hepatic steatosis[1]. Although overnutrition and obesity are closely associated with the development of NAFLD, protein deficiency and malnutrition have also been causally linked to NAFLD in humans and experimental models[2–4]. However, despite increasing research interest in understanding the diverse impact of individual amino acids have on obesity, insulin sensitivity and liver health[5–7], how dysregulation of hepatic amino acid metabolism on a background of NAFLD and dietary protein insufficiency promotes liver fat accumulation is still an enigma. Liver is a major metabolizing organ for the sulfur amino acids methionine, cysteine and taurine[8]. Hepatocytes acquire cysteine through dietary intake, proteolysis, the transsulfuration pathway, and glutathione (GSH) breakdown[9]. Methionine metabolism via the methionine cycle and transsulfuration pathway generates the methyl

[1]Harold Hamm Diabetes Center, Department of Physiology, University of Oklahoma Health Sciences Center, Oklahoma City, OK 73104, USA. [2]Department of Pharmacology, Toxicology and Therapeutics, University of Kansas Medical Center, Kansas City, KS 66160, USA. [3]Department of Cell Biology and Physiology, University of Kansas Medical Center, Kansas City, KS 66160, USA. [4]Laboratory for Molecular Biology and Cytometry Research, University of Oklahoma Health Sciences Center, Oklahoma City, OK 73104, USA. [5]Karsh Division of Gastroenterology and Hepatology, Department of Medicine, Cedars-Sinai Medical Center, Los Angeles, CA 90048, USA. ✉e-mail: tiangang-li@ouhsc.edu

donor S-adenosylmethionine (SAMe) and cysteine, which is used for the synthesis of the major antioxidant GSH and taurine that is primarily used for bile acid conjugation in the liver[7]. Analysis of human NAFLD and NASH samples revealed that approximately half of the patients showed serum metabolome reflecting dysregulated sulfur amino acid metabolism[10]. Genetic deletion of sulfur metabolizing genes resulted in progressive liver injury and cancer in mice[11–13]. However, the physiological regulation and pathogenic mechanisms associated with hepatic sulfur amino acid metabolism are still incompletely understood.

TFEB is a member of the basic helix-loop-helix leucine zipper family of transcription factors[14,15]. By activating a network of lysosomal and autophagy genes, TFEB acts as a nutrient and stress-sensing master regulator of lysosomal biogenesis and autophagy in various cell types and organ systems[16,17]. TFEB activity is mainly regulated by a cytosolic-nuclear shuttling mechanism whereby nutrient signaling phosphorylates TFEB to cause its cytosolic retention[17–20]. In response to nutrient deprivation and lysosomal stress TFEB is dephosphorylated and translocated into the nucleus where TFEB binds conserved E-box-like CLEAR DNA binding motif in its target gene promoters[16,17]. Activation of TFEB-mediated lysosomal clearance has been experimentally demonstrated to be beneficial in various forms of neurodegenerative diseases[21] and lysosomal storage diseases[22]. Recent studies have also uncovered a key role of TFEB in regulating various hepatic metabolic pathways, suggesting that TFEB activation is beneficial in chronic liver diseases[23–27]. By employing global metabolomics to gain new insights into TFEB regulation of hepatic metabolism, we have discovered a previously unrecognized role of TFEB in stimulating hepatic sulfur amino acid metabolic flux. Interestingly, this TFEB-mediated control of hepatic cysteine and GSH pool is found to act as a key driver of coenzyme A (CoA) production, which is subsequently shown to be critically required to support hepatic metabolic adaptation to lipid influx. Importantly, we find that this metabolic axis is selectively impaired in both fatty liver and dietary protein deficient conditions, which contributes to hepatic metabolic maladaptation and oxidative stress.

## Results

### Global metabolomics identified TFEB regulation of hepatic sulfur amino acid metabolism

In order to capture TFEB-induced early hepatic changes that contribute to its strong anti-NAFLD effect (Supplementary Fig. 1a-b)[23], we conducted global liver metabolomics in mice overexpressing hepatic TFEB and subsequently fed chow or challenged with Western diet (WD) for 1 week (Supplementary Fig. 1c-e). A total of 757 identified liver metabolites were assigned to super-pathways and sub-pathways. We next used a factor analysis-based filtering (1.5-fold cutoff, $q$ value < 0.1) followed by hierarchical clustering to reveal 9 distinct global patterns comprising of 429 metabolites based on TFEB and WD-dependent effects (Figs. 1a, b). Among these metabolites 43 were robustly upregulated by TFEB (>1.5-fold, $q$ value < 0.1) under both chow and WD conditions (Figs. 1c, d). Similarly, we identified 28 commonly downregulated metabolites upon TFEB overexpression independent of dietary conditions (Figs. 1c, e). Notably, these altered metabolites led to the identification of TFEB-induced sub-pathways (over-representation) including methionine, cysteine, SAMe and taurine metabolism, dipeptide metabolism, glutathione metabolism, and CoA metabolism that are closely linked to sulfur amino acid methionine and cysteine metabolism (Fig. 1d).

Upon further analysis, we found that TFEB significantly induced most intermediate metabolites in the methionine cycle and transsulfuration pathway (Figs. 2a, b), which also correlated with significantly enriched cysteine pool (Fig. 2b), GSH and the GSH breakdown metabolite cysteinylglycine (Fig. 2c). Liver γ-glutamylcysteine was increased by TFEB (Fig. 2d), suggesting increased de novo GSH synthesis.

However, TFEB does not increase glutamate cysteine ligase catalytic subunit (GCLC) or glutamate cysteine ligase modifier subunit (GCLM) (Fig. 2e), suggesting that increased GSH synthesis is likely driven by higher cellular cysteine availability. In addition, cysteine dioxygenase 1 (CDO1) protein was increased in TFEB overexpressing livers (Fig. 2e), which is consistent with the known effect of cysteine in feedforward stabilizing CDO1 protein to drive taurine synthesis (Fig. 2a)[8]. However, TFEB did not correspondingly increase hypotaurine or taurine (Fig. 2f). Since we reported that TFEB significantly increased hepatic bile acid synthesis and bile acid pool expansion[24], lack of hepatic taurine accumulation may be because newly synthesized taurine was incorporated into taurine-conjugated bile acids. Betaine and dimethylglycine (DMG) involved in homocysteine to methionine conversion were significantly reduced upon TFEB overexpression in chow-fed mice or upon WD feeding (Supplementary Fig. 2a). The mechanisms responsible for such changes are not fully clear. However, the ratio of betaine and DMG does not seem to be altered under these conditions. Sarcosine, which can be derived from DMG or glycine, was not induced by TFEB under chow condition (Supplementary Fig. 2b). Sarcosine was reduced upon WD feeding, which correlated with lower DMG but not glycine in these mice (Fig. 2g, Supplementary Fig. 2a). In polyamine synthesis pathway, we found that TFEB did not affect spermidine levels but increased the product 5′-methyl thio-adenosine (MTA) under chow diet condition that was also associated with modestly lower putrescine (Supplementary Fig. 2c), suggesting that higher SAMe may also promote the polyamine synthesis reactions. Taken together, the global metabolomics analysis provides a comprehensive view of how TFEB regulates the interconnected hepatic sulfur amino acid metabolism and its related pathways and reveals a previously unrecognized role of TFEB activation in stimulating sulfur amino acid metabolism leading to hepatic SAMe, cysteine and GSH enrichment.

### TFEB increases lysosomal proteolytic activity and methionine cycle-transsulfuration to maintain cellular cysteine and GSH pool

In addition to hepatic cysteine enrichment, we also observed that TFEB overexpression resulted in a general trend of increased amino acids despite being at a modest magnitude of ~10–20% compared to ~2-3-fold cysteine induction (Fig. 2g). TFEB is a strong inducer of autophagy–lysosome pathway[16,17]. Consistently, we found that the mRNA of autophagy genes and lysosomal genes *Ctsb* and *Ctsd*, encoding proteases cathepsin B and cathepsin D, respectively, was significantly induced upon TFEB overexpression (Supplementary Fig. 3a). Further, liver cathepsin B activity, a marker for lysosomal proteolytic activity, was significantly elevated in TFEB overexpressing livers (Supplementary Fig. 3b). Studies in livers showed that lysosomal proteolysis accounted for the majority of the protein degradation to maintain the intracellular amino acid pool especially during fasting and nutrient deprivation[28–31]. Studies with isolated liver lysosomes showed that lysosomal proteolysis generated not only free amino acids but also a significant amount of the dipeptides, which can be further hydrolyzed by dipeptidases to release amino acids[32–35]. In addition to elevated amino acids (Fig. 2g), we indeed found that TFEB caused a general increase of almost all dipeptides detected by metabolomics in mouse livers although the detected dipeptide species only accounted for a small portion of the cellular dipeptide pool (Supplementary Fig. 3c). Taken together, these findings provide additional evidence that TFEB induces hepatic lysosomal function to promote proteolysis. This mechanism may contribute to increased intracellular amino acids including methionine and cysteine but cannot fully explain the selectively enriched cysteine pool by TFEB (Fig. 2b).

Methionine metabolism through the methionine cycle- transsulfuration pathway is a major source of hepatic cysteine[7]. Given that many intermediates in this pathway are significantly increased by TFEB (Fig. 2b), we investigated the mechanisms mediating TFEB-

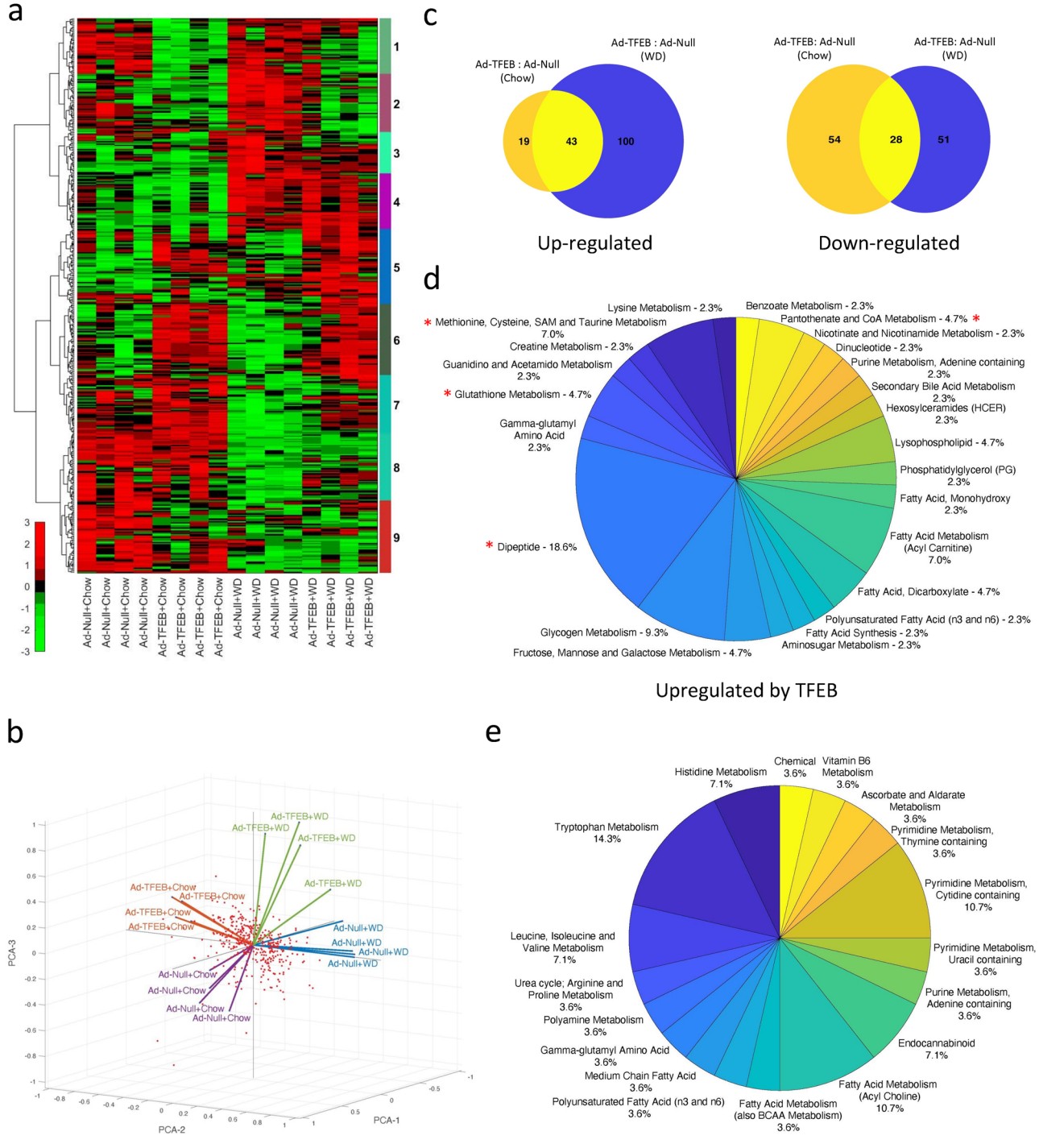

**Fig. 1 | Unbiased metabolomics identified TFEB regulation of hepatic sulfur amino acid metabolism pathways.** Male 10 weeks old C57BL/6 J mice were intravenously injected with Ad-Null or Ad-TFEB at $5 \times 10^8$ pfu/mouse. One week later, mice were fed chow (C) or Western diet (WD) for one additional week. Mice were fasted for 6 h and euthanized for tissue collection. Liver tissues were used for global metabolomics and bioinformatics analysis as described in the Method section ($n = 4$). **a** Heatmap of hierarchical clustering analysis. **b** Principal component analysis. **c** Number of upregulated or downregulated metabolites by TFEB overexpression independent of dietary conditions. **d, e** Over-represented pathways. A * indicates pathways of interest that were further investigated in this study.

activation of the transsulfuration pathway. We found that the mRNA of the liver-specific isoform methionine adenosyltransferase 1 A (MAT1A), which mediates the conversion of methionine to SAMe (Fig. 2a), was induced by TFEB under both chow and WD condition (Fig. 3a). TFEB induction of MAT1A was further confirmed in a separate cohort of chow-fed mice under fasting and refed condition (Supplementary Figs. 4a, b). Hepatic TFEB knockdown significantly decreased MAT1A mRNA by ~50% in mice (Figs. 3b, c). This result further supports that MAT1A is a TFEB transcriptional target, but it is possible that other members of the MiT/FTE family transcriptional factors may partially compensate for the loss of TFEB, therefore contributing to the modest reduction of MAT1A upon TFEB knockdown[36–38]. To further establish MAT1A as a TFEB target, we found that TFEB co-transfection strongly induced the luciferase

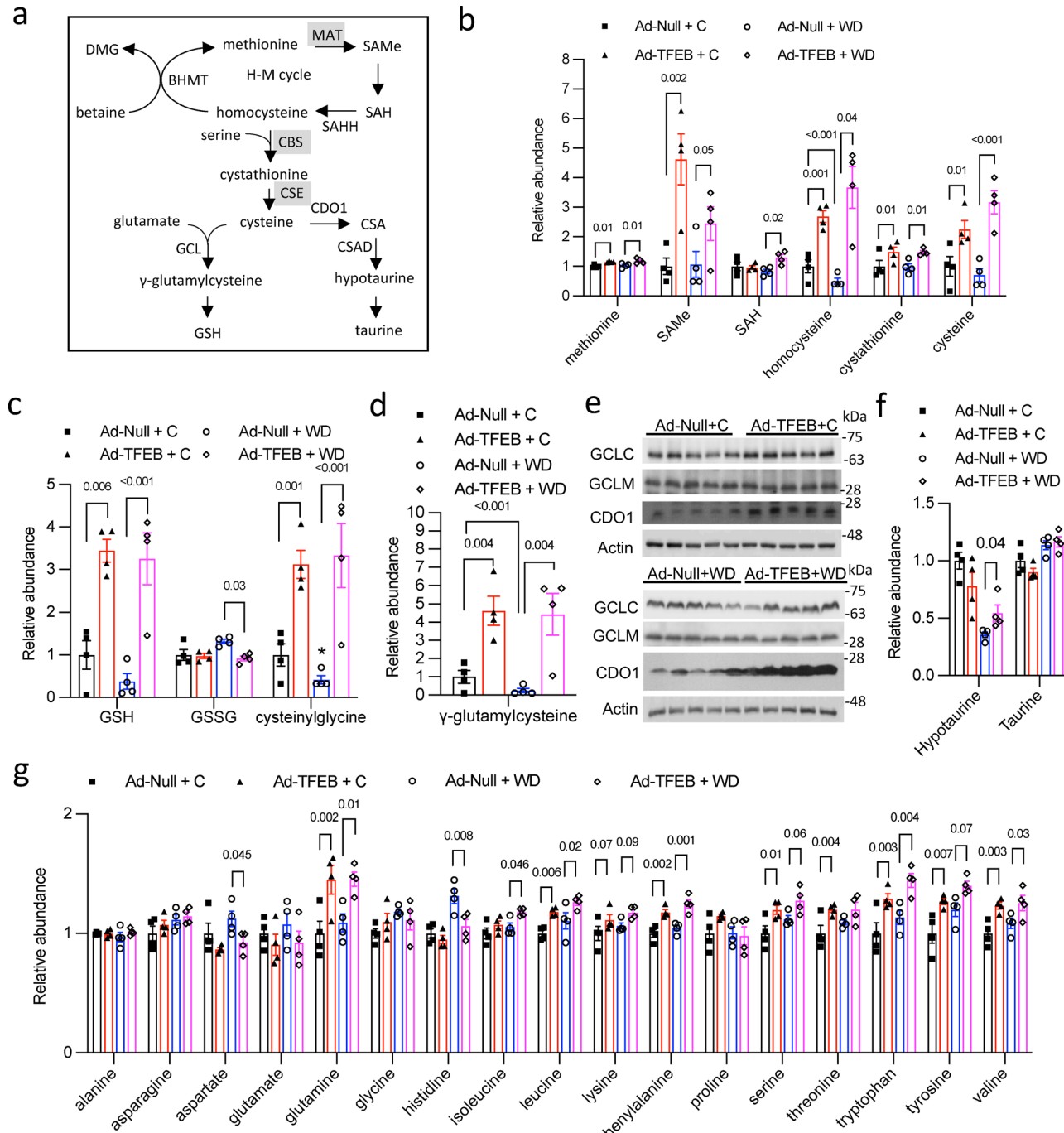

**Fig. 2 | TFEB stimulates methionine cycle-transsulfuration to increase hepatic cysteine and GSH pool. a** Illustration of methionine cycle-transsulfuration pathway and GSH and taurine synthesis. MAT1A Methionine adenosyltransferase 1A, CBS cystathionine β-synthase, CSE cystathionine γ lyase, SAMe S-adenosylmethionine, SAH S-adenosylhomocysteine, BHMT betaine-homocysteine S-methyltransferase, CDO1 cysteine dioxygenase 1, CSAD cysteine sulfinic acid decarboxylase, GCL glutamate cysteine ligase, GCLC glutamate cysteine ligase catalytic subunit, GCLM glutamate cysteine ligase modifier subunit, CSA cysteine sulfinic acid, H−M cycle homocysteine−methionine cycle, DMG dimethylglycine. GSH: reduced glutathione. **b**, **c**, **d** Male 10 weeks old C57BL/6 J mice were intravenously injected with Ad-Null or Ad-TFEB at 5×10⁸ pfu/mouse. One week later, mice were fed chow (C) or Western diet (WD) for one additional week. Relative abundance of liver metabolites is shown with control arbitrarily set as 1. Results are mean ± SEM (n = 4). **e** Western blot of liver protein expression in mice. Each band represents an individual mouse sample. **f, g** Male 10 weeks old C57BL/6 J mice were intravenously injected with Ad-Null or Ad-TFEB at 5×10⁸ pfu/mouse. One week later, mice were fed chow (C) or Western diet (WD) for one additional week. Relative abundance of liver metabolites is shown with control arbitrarily set as 1. Results are mean ± SEM (n = 4). Detailed statistical analysis of (**b**, **c**, **d**, **f**, **g**) is described under Metabolomics, statistical and bioinformatics analysis in the Methods section. A *p* value < 0.05 is considered statistically significant. Source data for (**b**, **c**, **d**, **e**, **f**, **g**) are provided as a Source Data file.

reporter activity driven by MAT1A promoter (Fig. 3d), suggesting a direct transcriptional activation of MAT1A by TFEB. Gene promoter sequence analysis revealed several putative E-box-like TFEB binding sites in the human and mouse MAT1A proximal promoters (Supplementary Table 1). Promoter DNA fragments containing each of the putative TFEB binding sites were then used to generate a series of luciferase reporter constructs. Luciferase reporter assays subsequently identified functional TFEB binding sites in human and

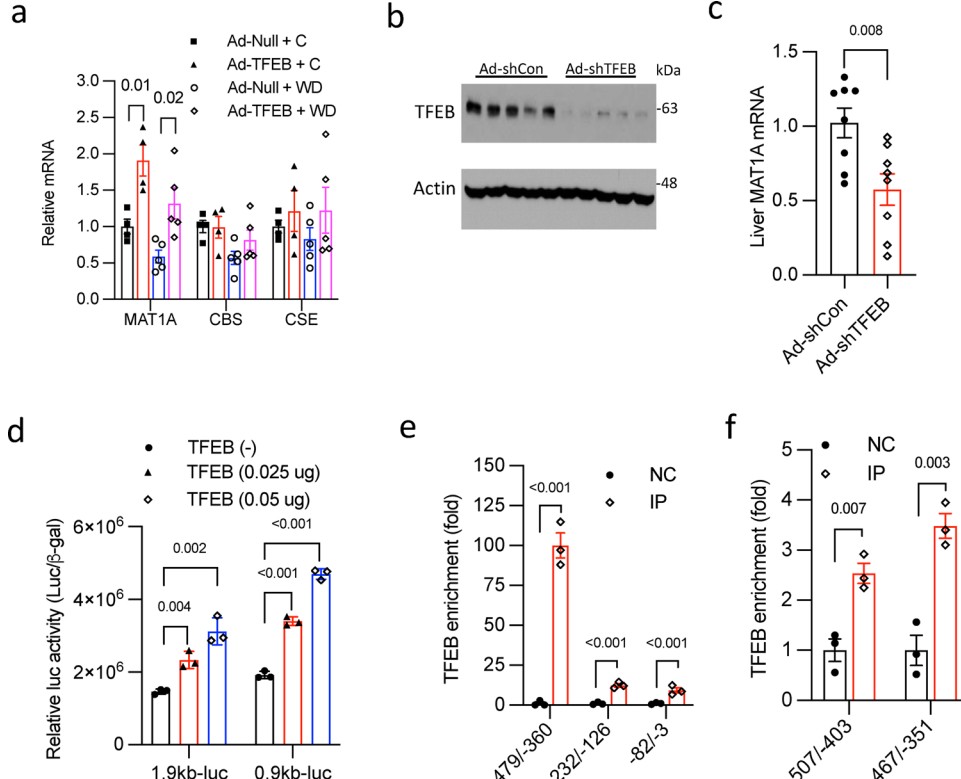

**Fig. 3 | TFEB stimulation of MAT1A transcription. a** Male 10 weeks old C57BL/6 J mice were intravenously injected with Ad-Null or Ad-TFEB at $5 \times 10^8$ pfu/mouse. One week later, mice were fed chow (C) or Western diet (WD) for one additional week. Relative liver mRNA are shown with control arbitrarily set as 1. (*n* = 5). MAT1A Methionine adenosyltransferase 1A, CBS cystathionine β-synthase, CSE cystathionine γ lyase. **b, c** Male 10 weeks old C57BL/6 J mice were intravenously injected with Ad-shCon (Ad-scramble) or Ad-shTFEB at $1 \times 10^9$ pfu/mouse. Mice were fed chow for 2 weeks and euthanized after 6 h fast (*n* = 8). Liver protein (**b**) and mRNA (**c**) were measured. Each band represents an individual mouse sample in (**b**). **d** Luciferase reporter constructs containing 1.9 kb or 0.9 kb human MAT1A promoter and TFEB expression plasmids were transfected in triplicates in AML12 cells

for 48 h. Luciferase activity was normalized to β-galactosidase activity. A representative of 2 independent experiments. **e, f** Chromatin immunoprecipitation assay with human liver nuclear fraction (**e**) or male 10 weeks old C57BL/6 J mouse liver nuclear fraction (**f**). NC negative control with IgG, IP immunoprecipitation with anti-TFEB antibody. Primer pair locations are relative to translational start site as +1. Results in a and c are expressed as mean ± SEM. Results in (**d, e,** and **f**) are expressed as mean ± SD of technical triplicate repeats. Two-way ANOVA and Tukey post hoc test are used for a and unpaired 2-tailed Student's *t*-test is used for (**c, d, e, f**). A *p* value < 0.05 is considered statistically significant. Source data for (**a–f**) are provided as a Source Data file.

mouse MAT1A promoters (Supplementary Figs. 4c, d). Consistently, chromatin-immunoprecipitation assay showed that TFEB was enriched in the human and mouse MAT1A proximal promoter regions (Figs. 3e, f). Downstream of the MAT1A, TFEB did not induce glycine N-methyltransferase (GNMT) (Supplementary Fig. 4e), which mediates the conversion of glycine to sarcosine using SAMe as a methyl donor. Cystathionine β-synthase (CBS) catalyzes the irreversible rate-limiting step of the transsulfuration pathway by converting homocysteine to cystathionine, which is further converted to cysteine by cystathionine γ-lyase (CSE) (Fig. 2a). TFEB overexpression did not affect CBS and CSE mRNA or protein expression (Fig. 3a, Supplementary Fig. 4e), suggesting that TFEB may not directly induce transsulfuration pathway. Taken together, these results suggest a mechanism whereby TFEB induces hepatic autophagy and lysosome function to generate intracellular amino acids and further stimulates methionine cycle and synthesis of SAMe, which promotes downstream transsulfuration pathway to selectively enrich hepatic cysteine and GSH pool.

## TFEB induces hepatic CoA synthesis to support fatty acid metabolism

Interestingly, the top TFEB-induced sub-pathways also include synthesis of CoA (Fig. 1d), which is an essential cofactor participating in a large number of biological pathways including anabolic/catabolic metabolism[39]. The majority of the hepatic CoA pool (~75%)

is found in the mitochondria[40], and one of the most important functions of CoA is to form acyl-CoA to support mitochondrial fatty acid β-oxidation[41]. Further analysis of the metabolites in this pathway revealed that WD challenge rapidly decreased CoA and CoA synthesis intermediate metabolites, which were significantly induced by TFEB under chow and WD conditions (Figs. 4a, b). Despite the absence of hepatic steatosis after 1-week WD challenge (Supplementary Figs. 5a, b), hepatic β-hydroxybutyrate, a marker of fatty acid oxidation, was higher in TFEB overexpression groups (Fig. 4c). The effect of hepatic CoA synthesis defect in promoting hepatic steatosis has been previously demonstrated in liver *Pank1* knockout mice[41]. Consistently, we show that knockdown of hepatic PANK1 resulted in ~40% reduction of hepatic CoA (Supplementary Figs. 6a, b), which was similar to that reported in *Pank1* knockout mice[41]. Upon 2 weeks WD challenge, hepatic PANK1-deficient mice developed worsened steatosis with ~2-fold higher hepatic triglyceride (TG) and cholesterol content than controls, which could be prevented by TFEB overexpression (Figs. 4d, e, f). Furthermore, TFEB overexpression did not transcriptionally induce PANK1 mRNA in mouse livers (Supplementary Figs. 6c, d). Taken together, we demonstrate that hepatic CoA availability plays an important role in supporting hepatic metabolic adaptation to lipid overload and preventing steatosis. TFEB induces hepatic CoA synthesis to support hepatic fatty acid oxidation. However, this TFEB effect is not mediated through TFEB regulation of PANK1 expression.

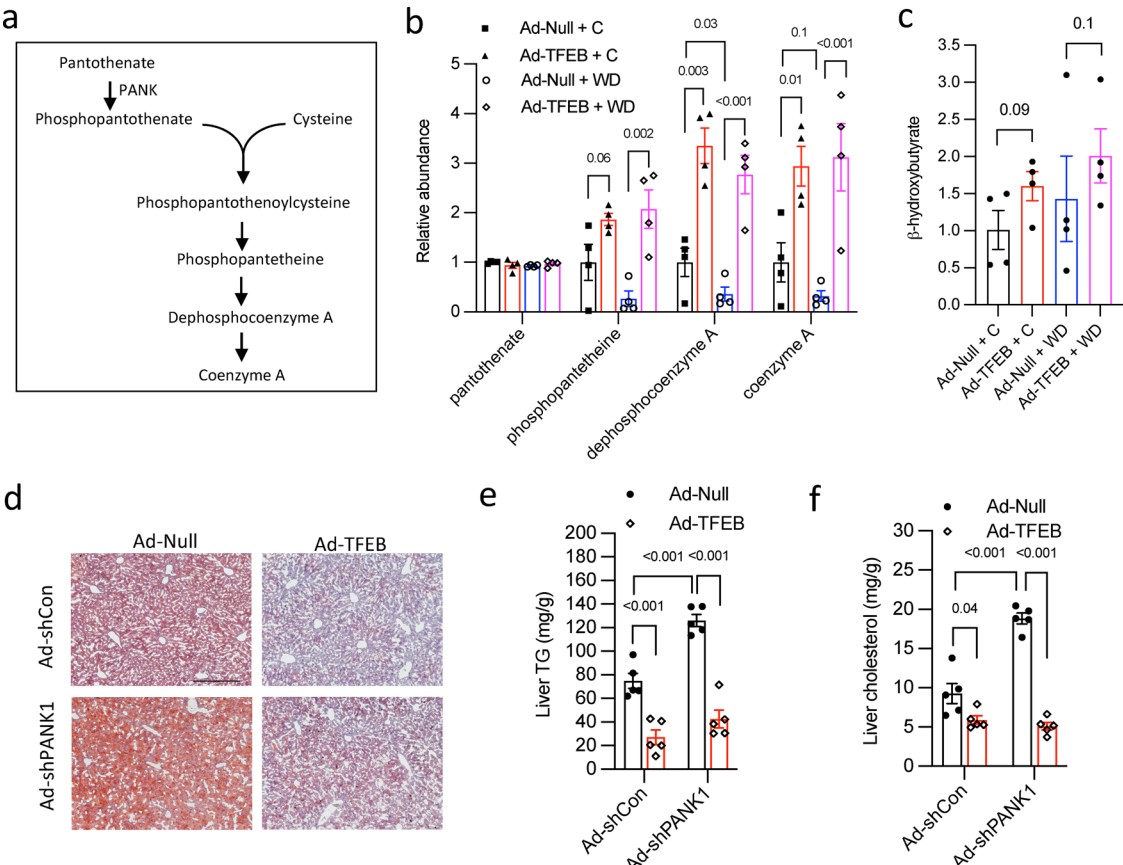

**Fig. 4 | TFEB stimulates hepatic CoA synthesis to support fatty acid metabolism. a** Illustration of the Coenzyme A (CoA) synthesis pathway. PANK: patothenate kinase. **b, c** Male 10 weeks old C57BL/6 J mice were intravenously injected with Ad-Null or Ad-TFEB at $5 \times 10^8$ pfu/mouse. One week later, mice were fed chow (C) or Western diet (WD) for one additional week. Mice were fasted for 6 h and euthanized. Relative abundance of liver metabolites are shown with control arbitrarily set as 1 ($n = 4$). **d, e, f** Adenovirus injected male 10 weeks old C57BL/6 J mice (Ad-Null and Ad-TFEB at $5 \times 10^8$ pfu/mouse and Ad-shCon (Ad-scramble) and Ad-shPANK1 at

$1 \times 10^9$ pfu/mouse) were fed WD for 2 weeks and euthanized after 6 h fast ($n = 5$). **d** Representative liver Oil Red O stain. Scale bar = 250 μm. **e** Liver triglycerides (TG). **f** Liver cholesterol (Chol). Results in (**b, c, e, f**) are presented as mean ± SEM. Detailed statistical analysis of (**b**) and (**c**) is described under Metabolomics, statistical and bioinformatics analysis in the Methods section, and 2-way ANOVA and Tukey post hoc test are used for **e** and **f**. A $p$ value < 0.05 is considered statistically significant. Source data for (**b, c, e, f**) are provided as a Source Data file.

## Increased cellular cysteine availability mediates TFEB-driven hepatic CoA synthesis

Cells cannot directly uptake CoA and thus acquire CoA solely via a well conserved de novo synthesis pathway using pantothenate (vitamin B5) and cysteine as substrates[39]. Despite the key role of CoA in supporting cellular metabolism, the regulation of hepatic CoA synthesis is poorly defined. Among the two CoA synthesis substrates, TFEB-induction of CoA correlated with increased hepatic cysteine but not pantothenate (Fig. 2b, Fig. 4b). Indeed, all CoA synthesis intermediates downstream of the cysteine addition step were upregulated by TFEB (Fig. 4b). This prompted us to hypothesize that TFEB increases cysteine availability to stimulate CoA synthesis via a substrate-driven mechanism. If true, experimental manipulation that decreases hepatic cysteine pool should reduce CoA synthesis. Because transsulfuration, lysosomal proteolysis and dietary protein are three important sources of cellular cysteine pool, we next tested our hypothesis using in vivo and in vitro models with the manipulation of each of the three pathways.

We first knocked down hepatic MAT1A to decrease cysteine synthesis from methionine (Fig. 5a, Supplementary Fig. 7a)[11]. An ~80% reduction of hepatic MAT1A is associated with ~2-fold higher methionine but only modestly decreased hepatic cysteine (~20%, $p = 0.2$) (Figs. 5b, c). Given that GSH is a major cellular cysteine reservoir, we further measured GSH and its breakdown metabolite cysteinylglycine that can be hydrolyzed to release cysteine into the intracellular pool.

We found that both GSH and cysteinylglycine were ~30% lower in MAT1A deficient livers (Fig. 5d), which supports decreased hepatic overall cysteine availability. Hepatic MAT1A deficient mice show slightly reduced CoA ($p = 0.17$) but ~50% lower long chain acyl-CoA species (Fig. 5e), which reflects a decreased hepatic CoA pool. Despite the absence of steatosis under chow diet condition (Supplementary Fig. 7b), there is an overall trend of elevated hepatic medium chain acylcarnitines and significantly decreased β-hydroxybutyrate (Supplementary Fig. 7c, Fig. 5f), suggesting that hepatic MAT1A deficiency is associated with reduced hepatic fatty acid oxidation. Indeed, when challenged with WD, hepatic MAT1A deficient mice accumulated ~40% more TG, ~60% more cholesterol and showed worsened steatosis than controls (Figs. 5g, h, i). Worsened hepatic steatosis was also associated with increased hepatic very low-density lipoprotein (VLDL) -TG secretion in MAT1A deficient mice (Supplementary Fig. 7d). Furthermore, hepatic MAT1A deficient mice also showed elevated transaminases upon WD challenge (Fig. 5j). To further determine the relative importance of MAT1A induction in mediating TFEB stimulation of cysteine and CoA synthesis, we next compared the TFEB regulation of these metabolites in control and liver MAT1A deficient mice that were fasted for 16 h to minimize the dietary source of sulfur amino acids (Fig. 5k). We found that blocking TFEB induction of MAT1A significantly prevented TFEB induction of hepatic cysteine (Fig. 5l), GSH (Supplementary Fig. 7e), and γ-glutamylcysteine, the marker of de

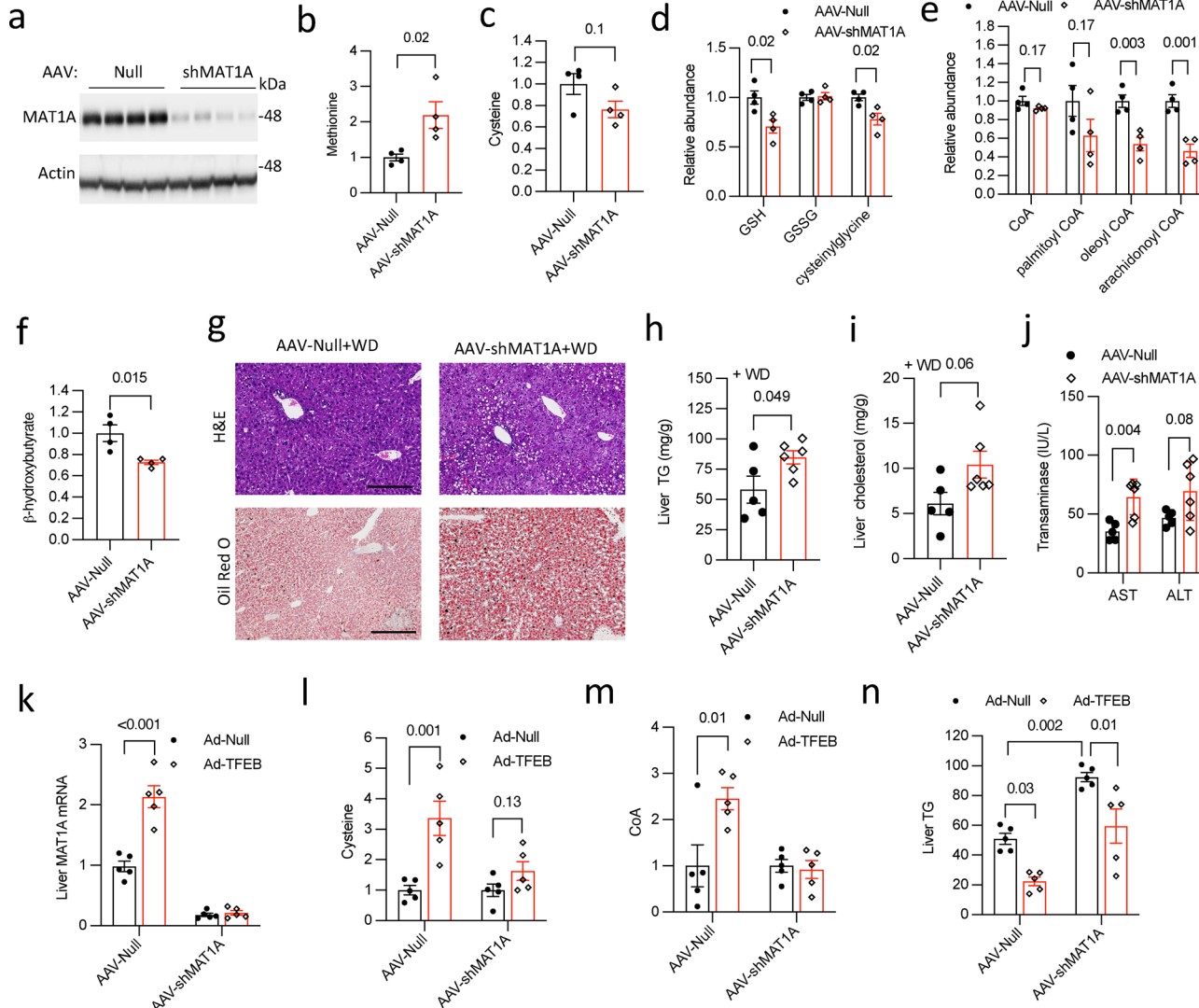

**Fig. 5 | Hepatic MAT1A deficient mice have reduced hepatic cysteine and CoA pool and higher susceptibility to developing steatosis and liver injury when challenged with Western diet. a–f** Male 10 weeks old C57BL/6 J mice were intravenously injected with AAV-Null or AAV-shMAT1A at $2 \times 10^{11}$ GC/mouse. Mice were maintained on chow diet for 2 weeks and euthanized after 6 h fast ($n = 4$). **a** Liver methionine adenosyltransferase 1 A (MAT1A) protein. Each band represents an individual mouse sample. **b, c, d, e, f** Relative abundance of liver metabolites is shown with control arbitrarily set as 1. GSH: reduced glutathione; CoA: coenzyme A. **g, h, i, j** Male 10 weeks old C57BL/6 J mice were intravenously injected with AAV-Null or AAV-shMAT1A at $2 \times 10^{11}$ GC/mouse. Mice were then fed Western diet (WD) for 6 weeks and euthanized after 6 h fast ($n = 5$ for AAV-Null, $n = 6$ for AAV-shMAT1A).

**g** Representative liver H&E and Oil red O stain. Scale bar = 250 μm. **h** Liver triglycerides (TG). **i** Liver cholesterol. **j** Serum aspartate amino transferase (AST) and alanine amino transferase (ALT). **k, l, m, n**. Male 10 weeks old C57BL/6 J mice were intravenously injected with AAV-Null or AAV-shMAT1A at $2 \times 10^{11}$ GC/mouse, and Ad-Null or Ad-TFEB ($5 \times 10^{8}$ pfu/mouse) as indicated. Mice were then fed WD for 2 weeks and euthanized after 16 h fast. $n = 5$. **k** Liver mRNA. **l, m** Relative abundance of liver cysteine and CoA is shown with respective control arbitrarily set as 1. **n** Liver TG. All results are mean ± SEM. A $p$ value < 0.05 is considered statistically significant (Unpaired 2-tailed $t$-test for (**b, c, d, e, f, h, i, j**); 2-way ANOVA and Tukey post hoc test for **k, l, m, n**). Source data for (**a, b, c, d, e, f, h, i, j, k, l, m, n**) are provided as a Source Data file.

novo GSH synthesis (Supplementary Fig. 7f). Indeed, TFEB induction of liver CoA was prevented when MAT1A induction was abolished (Fig. 5m). Under AAV-Null condition, TFEB reduced liver TG by ~60%. The magnitude of TFEB-mediated hepatic TG reduction was attenuated but not fully abolished upon MAT1A knockdown (Fig. 5n, Supplementary Fig. 7g), which is expected since TFEB is known to lower hepatic steatosis via other mechanisms[23]. Taken together, these results suggest that MAT1A deficiency contributes to hepatic cysteine and CoA deficiency and susceptibility to hepatic steatosis, and selective hepatic cysteine enrichment and stimulation of CoA synthesis are largely mediated by TFEB induction of hepatic MAT1A.

We next ask if the autophagy−lysosome pathway also serves as a significant source of methionine and cysteine, among other amino acids, to feed into the downstream CoA synthesis. To answer this question, we directly inhibited the autophagy−lysosome pathway in AML12 cells with chloroquine (CQ) and studied its impact on cellular sulfur amino acid and CoA abundance (Supplementary Fig. 8a). In cells cultured in the high end of the physiological range of methionine and cystine (~200 μM), CQ caused significant but only modest reduction of cellular methionine, cystine, GSH and CoA (Supplementary Figs. 8b, c, d, e). When cells were cultured in methionine and cystine free medium for 6 h during which cells relied solely on the endogenously generated methionine and cysteine, these metabolites were all rapidly reduced by ~50−60%. Under this condition, CQ treatment caused further reduction of these metabolites of 50% or more (Supplementary Figs. 8b, c, d, e). These results suggest that when exogenous source of amino acids is limited, lysosomes indeed play an important role in maintaining cellular methionine and cysteine pool. Therefore, TFEB

activation of the autophagy–lysosome pathway in response to fasting or nutrient deprivation may also contribute to the cellular cysteine availability that supports CoA synthesis.

**Dietary protein insufficiency causes selective hepatic cysteine deficiency that impairs CoA synthesis and fatty acid oxidation**

The findings from the MAT1A deficient model support the role of the TFEB-MAT1A axis in maintaining hepatic cysteine/GSH and CoA pool to support hepatic metabolic adaptation. However, the relatively modest reduction of hepatic cysteine and GSH pool in hepatic MAT1A deficient mice suggest that dietary protein intake is also a significant hepatic cysteine source, which prompted us to further test the hepatic cysteine-CoA link via dietary manipulation. To this end, we first generated a 5% casein WD by decreasing the total protein source casein in WD from 20% to 5% (Supplementary Table 2). We then generated a 5% casein+cys WD by adding cystine to restore total dietary sulfur amino acids to the level in 20% casein WD. Corn starch was used to adjust the energy density so that three diets are isocaloric. After a relatively short period of 6 weeks feeding, all three groups showed similar body weight and had not developed obesity at the time of analysis (Supplementary Fig. 9a). Metabolomics analysis revealed that feeding mice the 5% casein WD significantly decreased hepatic cysteine by ~90%, which was restored by cystine supplementation to ~60% of that in 20% casein WD-fed mice (Fig. 6a). However, it was intriguing to note that hepatic cysteine pool was uniquely sensitive to dietary protein restriction because other amino acids were either unaltered, only modestly reduced, or even increased in mice fed the 5% casein diet (Supplementary Fig. 9b). Interestingly, many γ-glutamyl amino acids were increased upon 5% casein diet feeding and decreased after cystine supplementation (Supplementary Fig. 9c, d). This result suggests a yet to be identified cysteine sensing mechanism that induces the γ-glutamyl cycle, which may act as an adaptive mechanism to increase extracellular amino acid import and prevent a general decrease of cellular amino acid pool. However, this appears insufficient to maintain intracellular cysteine pool, possibly due to the higher demand for cysteine as the precursor of GSH, taurine, and CoA that are usually maintained at mM concentration in the liver[8,9,42–44]. In cultured cells, although amino acid starvation is a strong inducer of autophagy, sulfur amino acid deficiency, per se, did not affect autophagy flux (Supplementary Fig. 8a). Liver methionine, SAMe and cystathionine were largely unaffected by cystine supplementation (Fig. 6g), which is expected given that transsulfuration is irreversible. Hepatic MAT1A, GNMT, CBS and nuclear TFEB protein were also similar among the three groups (Supplementary Figs. 9e, f). Other mechanisms that potentially contribute to selective hepatic cysteine deficiency in response to dietary protein restriction likely exist but are out of the scope of this study. In close correlation with the cellular cysteine availability, γ-glutamylcysteine, GSH, GSSG, GSH breakdown product cysteinylglycine, and taurine were markedly reduced in mice fed 5% casein diet, which was fully restored upon cystine supplementation (Fig. 6b–f), supporting that cellular cysteine availability dictates downstream sulfur amino acid metabolic flux.

Consistent with reduced hepatic cysteine availability, mice fed the 5% casein diet did not have altered hepatic pantothenate level but showed decreased CoA as well as CoA synthesis intermediates downstream of the cysteine addition step (Fig. 6h). The finding that dietary cystine supplementation without normalization of the total dietary protein content fully restored hepatic CoA and CoA synthesis intermediates supported our hypothesis that cysteine availability was a determinant of hepatic CoA synthesis. Interestingly, mice fed the 5% casein diet showed marked hepatic accumulation of almost all detected acylcarnitines species (Fig. 6i), a hallmark feature of hepatic CoA insufficiency regardless of underlying causes[41,45]. Notably, hepatic accumulation of acylcarnitines was largely reversed by dietary cystine supplementation that also restored hepatic cysteine and CoA synthesis

(Figs. 6a, h, i). Hepatic long-chain fatty acids were found to be similar among the 3 groups, suggesting that acylcarnitine accumulation was not likely driven by increased substrate concentration (Supplementary Fig. 10). Mice fed a 5% casein diet showed increased steatosis with uniformly pale hepatocytes and increased Oil Red O stain confirming neutral lipid accumulation. Interestingly, cystine supplementation significantly prevented lipid droplet accumulation in the pericentral areas with remaining lipid droplet accumulation remained visible surrounding the periportal areas (Supplementary Fig. 11), suggesting that dietary protein deficiency also promotes hepatic lipid accumulation in a zonal manner.

Given the potential confounding systemic effect of dietary protein deficiency on organ metabolism, we next performed Seahorse XF palmitate oxidation stress test in cultured liver cells that were cultured in regular medium with ~200 μM sulfur amino acids or exposed to low end of the physiological range of sulfur amino acids (20 μM)[8]. Low level of sulfur amino acid exposure did not alter basal $O_2$ consumption rate (OCR) but significantly impaired maximal palmitate-dependent OCR under saturating palmitate and L-carnitine concentration (Fig. 6j). Consistently, cells exposed to low level of sulfur amino acid showed significantly higher intracellular lipid droplet accumulation (Fig. 6k). These findings further solidify our in vivo observation that cysteine deficiency indeed limits fatty acid oxidation capacity in a cell autonomous manner. Collectively, these studies revealed that the selective hepatic cysteine deficiency, but rather overall amino acid deficiency, was the underlying cause of marked acylcarnitine accumulation under the condition of dietary protein deficiency, which provided a molecular mechanism linking dietary protein deficiency to the development of hepatic steatosis in humans and rodent models of dietary protein insufficiency[2,3].

**Fatty liver is a cysteine and CoA insufficient state that contributes to metabolic maladaptation**

To obtain additional insights on hepatic sulfur amino acid and CoA metabolism in NAFLD, we performed metabolomics analysis of livers from chow and 16 weeks WD fed mice with profound hepatic steatosis (Fig. 7a). Hepatic methionine, S-adenosylhomocysteine, homocysteine, cystathionine and cysteine were significantly decreased in fatty livers than controls (Fig. 7b), indicating reduced hepatic sulfur amino acid flux in NAFLD. The protein levels of MAT1A, GNMT and CBS were also lower by various magnitudes in fatty livers than controls (Fig. 7c). Furthermore, hepatic CoA and CoA synthesis intermediates except pantothenate were all significantly reduced in 16-wk WD-fed mice (Fig. 7d), which is consistent with the close link of hepatic cysteine pool and CoA synthesis. It is noted that chronic WD-fed mice showed elevated hepatic β-hydroxybutyrate ($p = 0.07$) (Fig. 7e), indicating higher overall fatty acid oxidation possibly due to increased fatty acid abundance (Fig. 7f). A few detected acyl-CoA species do not show a uniform pattern of change in fatty livers (Fig. 7g). However, fatty livers accumulated many medium chain and long chain acylcarnitines but not short chain acylcarnitines (Fig. 7h), which reflects the hepatic CoA insufficient condition[41,45]. Taken together, these results suggest that cysteine, GSH and CoA insufficiency renders fatty livers maladaptive to excessive fatty acid influx, which, in the presence of elevated oxidative stress, contributes to accumulation of toxic lipid intermediates that exacerbate liver inflammation and injury.

## Discussion

Using a global metabolomics approach, we identified a role of TFEB in the regulation of hepatic sulfur amino acid metabolism, which in turn supports hepatic lipid catabolism and prevents oxidative injury[23,24,46]. As illustrated in Fig. 8, our findings suggest a model whereby TFEB activation stimulates autophagy-lysosomal proteolysis and methionine cycle-transsulfuration pathway resulting in hepatic enrichment of cysteine that drives downstream production of GSH, taurine, and CoA.

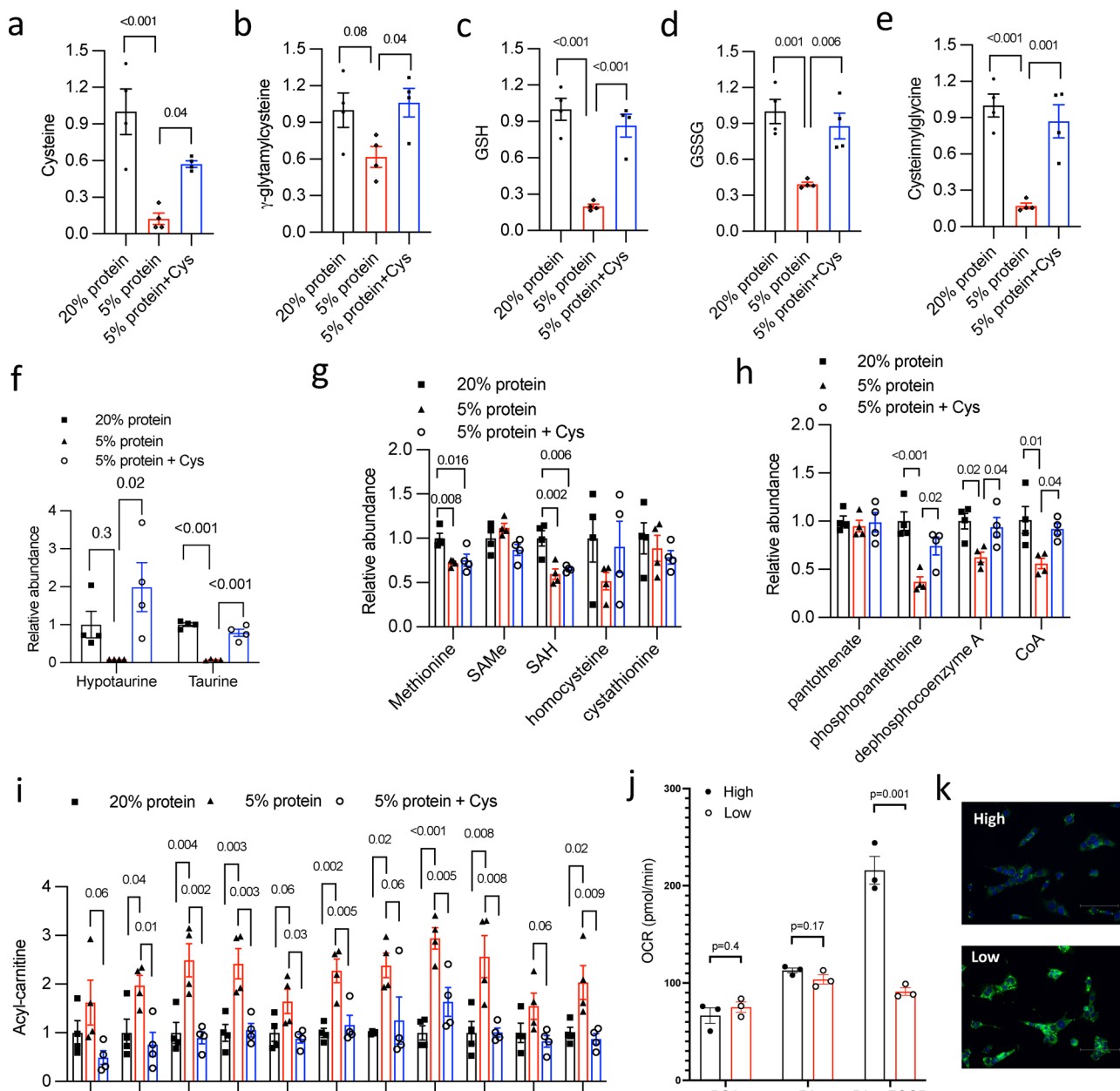

**Fig. 6 | Hepatic cysteine availability is a key determinant of hepatic CoA synthesis and maximal fatty acid oxidation capacity. a–i** Male 10 weeks old C57BL/6 J mice were fed isocaloric Western diet containing 20% protein, 5% protein or 5% protein with added cystine (Cys) for 6 weeks. Mice were euthanized after 6 h fast. Liver tissues were used for metabolomics analysis (*n* = 4). Relative abundance of liver metabolites is shown with control arbitrarily set as 1. All results are mean ± SEM. A *p* value < 0.05 is considered statistically significant (One-way ANOVA and post hoc Dunnett test). GSH reduced glutathione, SAMs S-adenosylmethionine, SAH S-adenosylhomocysteine, CoA coenzyme A. **j.** AML12 cells were cultured in medium containing 200 μM (High) or 20 μM (Low) methionine and cystine overnight. Seahorse XF palmitate oxidation stress test was performed in live cells under basal and FCCP-induced mitochondrial uncoupled conditions as described in Methods. Results are expressed as mean ± SD (technical repeats). The results are from one of two independent experiments. Unpaired 2-tailed *t*-test was used to calculate the *p* values. BSA bovine serum albumin, PA palmitate (complexed to BSA). **k.** AML12 cells were cultured in medium containing 200 μM (High) or 20 μM (Low) methionine and cystine overnight. Lipid droplets were visualized with BODIPY staining and nuclei were visualized with DAPI staining. Representative fluorescence images from 3 independent experiments are shown. Scale bar: 125 μm. Source data for (**a–j**) are provided as a Source Data file.

The critical roles of these intermediate metabolites in regulating cellular redox homeostasis and lipid and bile acid metabolism have been well documented in various liver disease conditions[7,9,47]. GSH deficiency, primarily owing to lower de novo synthesis, is commonly associated with aging, diabetes, and chronic liver diseases[9,48–51]. Another key finding of our study is that TFEB-mediated cysteine enrichment also acts as a driver of CoA production, which so far has poorly defined physiological regulation. This finding is consistent with TFEB being a nutrient sensing transcriptional factor that promotes

hepatic lipid catabolism[23]. As a major fatty acid metabolizing organ, the importance for the liver to maintain a sufficient CoA pool has been substantiated in this study and previously[41,45]. These findings collectively support the pathophysiological significance of TFEB-mediated regulation of hepatic sulfur amino acid metabolism.

Our study suggests that a few mechanisms may act coordinately to mediate the TFEB effect on hepatic cysteine pool. During starvation, hepatocytes primarily rely on autophagy-lysosomal proteolysis as a major mechanism to maintain intracellular amino acid pool[52]. We show

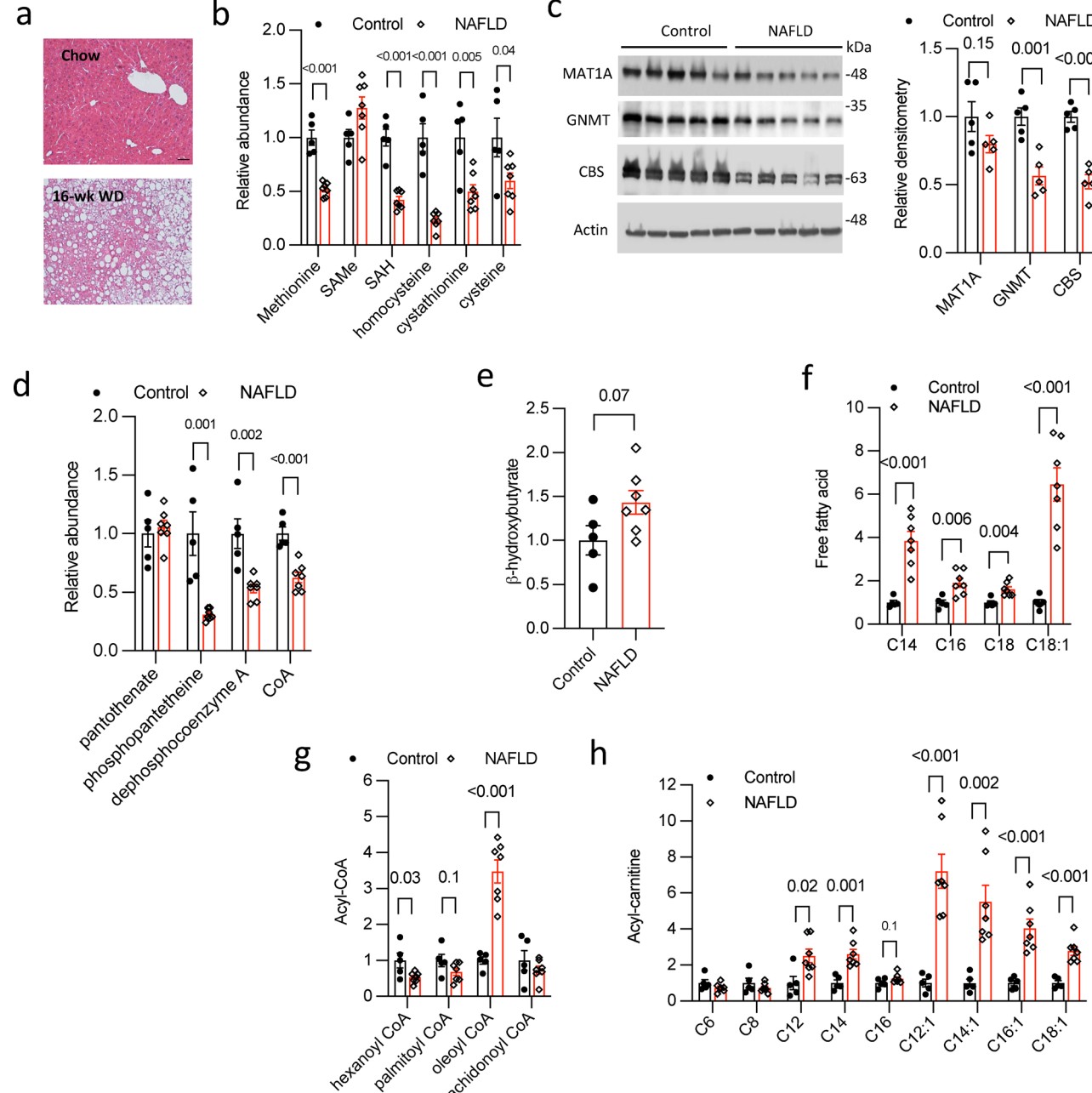

**Fig. 7 | NAFLD is a cysteine and CoA insufficient state that contributes to hepatic metabolic inflexibility. a–h** Male 10 weeks old C57BL/6 J mice were fed chow diet (Control) or Western diet (WD) for 16 weeks to induce non-alcoholic fatty liver disease (NAFLD). Mice were fasted for 6 h and euthanized. (*n* = 5 for Control, *n* = 7 for NAFLD). **a** Representative liver H&E stain. Scale bar = 50 μm. **b**, **d**, **e**, **f**, **g**, **h** Liver tissues were used for metabolomics analysis. Relative abundance of liver metabolites is shown with control arbitrarily set as 1. **c** Left panel: Western blots of liver proteins. Each band represents an individual mouse sample. Right panel: relative densitometry normalized to Actin band intensity. Densitometry is determined by ImageJ software. All results are mean ± SEM. A *p* value < 0.05 is considered statistically significant (Unpaired 2-tailed *t*-test). Source data for (**b**, **c**, **d**, **e**, **f**, **g**, **h**) are provided as a Source Data file. SAMe S-adenosylmethionine, SAH S-adenosylhomocysteine, MAT1A Methionine adenosyltransferase 1A, GNMT Glycine N-Methyltransferase, CBS cystathionine β-synthase, CoA coenzyme A.

that this mechanism is activated by TFEB to contribute to the generally elevated amino acids. However, this mechanism alone does not fully explain the selective enrichment of cysteine in TFEB overexpressing livers, which is shown to be mainly attributed to TFEB-mediated induction of MAT1A. It is estimated that ~50% of the intracellular GSH pool in hepatocytes can be derived from homocysteine[53], supporting the importance of the transsulfuration pathway in intracellular cysteine generation. Another mechanism that likely further contributed to hepatic cysteine enrichment is elevated GSH pool in TFEB overexpressing livers. To prevent intracellular cysteine accumulation,

increased cysteine availability promotes cysteine incorporation into the GSH pool[9], and, via stabilizing CDO1 protein, stimulates the synthesis of taurine that is used for bile acid conjugation[8]. Although conversion of cysteine to taurine is irreversible, GSH, being the most abundant cellular thiol carrier, acts as a major cysteine reservoir. In response to fasting, hepatic cysteine pool is primarily maintained at the expense of GSH breakdown. Therefore, higher GSH pool may further contribute to a larger steady state cysteine pool (Fig. 8).

Despite the well-established role of CoA in supporting hepatic lipid metabolism[41,45], the regulation of hepatic CoA synthesis under

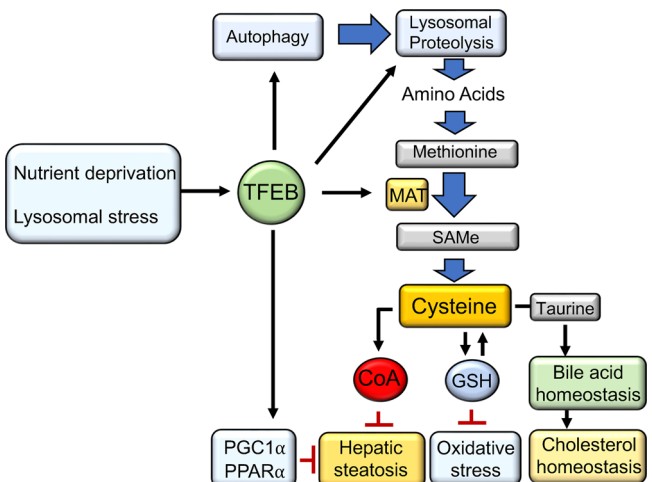

**Fig. 8 | TFEB regulation of hepatic lipid metabolism and antioxidant defense.**
TFEB is a nutrient and stress-sensing transcriptional factor that plays an important role in regulating cellular metabolic homeostasis. This study shows that TFEB, via activation of the hepatic autophagy-lysosomal proteolysis and induction of sulfur amino acid flux, increases hepatic cysteine availability that drives downstream CoA, GSH and taurine synthesis. In addition, previous studies show that TFEB, partly via PGC1α induction, promotes PPARα activation and fatty acid oxidation. Under NAFLD condition, incomplete fatty acid oxidation produces ROS to increase oxidative stress and mitochondrial dysfunction. This study shows that TFEB mediates the synthesis of GSH that enhances hepatic antioxidant defense and redox homeostasis. Previous study reports that TFEB induces bile acid synthesis to lower cholesterol. TFEB-mediated cysteine enrichment also drives the synthesis of taurine, which supports bile acid conjugation and homeostasis. In summary, this study reveals TFEB regulation of hepatic sulfur amino acid metabolism that links TFEB to control of hepatic metabolic and redox homeostasis. SAMe S-adenosylmethionine, MAT Methionine adenosyltransferase, CoA coenzyme A, GSH reduced glutathione, PPARα peroxisome proliferator-activated receptor α, PGC1α PPARγ coactivator α.

pathophysiological conditions is poorly defined. Our study showed that TFEB significantly increased hepatic CoA synthesis through a cysteine driven mechanism. The cysteine-CoA link is subsequently demonstrated in several in vitro and in vivo models with inhibited cysteine input pathways, including transsulfuration, dietary protein intake, and autophagy–lysosome proteolysis. Hepatic MAT1A knockdown results in reduction of hepatic cysteine/GSH pool which correlates with lower CoA/acyl-CoA pool and reduced fatty acid oxidation marker β-hydroxybutyrate. Consistently, upon WD challenge, hepatic MAT1A deficient mice showed worsened steatosis and higher liver injury. It should be noted that MAT1A deficiency has been linked to increased liver steatosis and injury via other mechanisms in addition to reduced liver CoA. Notably, MAT1A deficiency has been reported to alter hepatic VLDL-TG secretion[54]. MAT1A knockout mice at young age show altered phospholipid composition and reduced hepatic VLDL-TG secretion to contribute to intrahepatic lipid accumulation. After MAT1A knockout mice develop spontaneous hepatic steatosis at an older age, their hepatic VLDL-TG significantly increases, which resembles our findings in liver MAT1A knockdown mice with worsened hepatic steatosis. Using dietary protein manipulation approaches, we unexpectedly found that dietary protein deficiency resulted in a more profound hepatic cysteine deficient phenotype without an overall hepatic amino acid deficiency, possibly due to hepatic adaptive responses to dietary protein deficiency. Using this model, we were able to further demonstrate that hepatic cysteine deficiency not only reduced CoA synthesis, but also caused marked accumulation of acylcarnitines, a characteristic feature of hepatic CoA deficiency[41,45]. Notably, dietary cystine supplementation that increased hepatic cysteine abundance restored hepatic CoA synthesis and completely reversed hepatic acylcarnitine accumulation without the need to

normalize total dietary protein intake, suggesting that impaired hepatic cysteine-CoA axis may underlie the dietary protein deficiency associated hepatic mitochondrial dysfunction and NAFLD development[2,3]. It seems paradoxical that during fasting the dietary cysteine source is diminished but hepatic CoA demand is increased in response to higher fatty acid influx. Therefore, the autophagy-lysosomal proteolysis, GSH breakdown and methionine cycle-transsulfuration pathway become more important in maintaining cellular cysteine and CoA pool when dietary protein source is limited. This is further supported by more direct evidence from cells treated with lysosome inhibitor CQ, which caused a rapid and significant drop in cellular cysteine and CoA concentration when exogenous sulfur amino acid supply was removed. However, reduced autophagic flux, lysosomal dysfunction, impaired TFEB activation, and dysregulation of sulfur amino acid metabolism have all been documented in experimental and human fatty livers[10,25,46,55,56]. We further show that reduced hepatic transsulfuration metabolites correlate with impaired CoA synthesis and hepatic acylcarnitine accumulation in fatty livers despite overall increased fatty acid oxidation as indicated by higher β-hydroxybutyrate, which collectively reflect a CoA insufficient state. These metabolic defects could be a combined result of decreased cysteine pool and increased utilization of GSH and CoA in fatty livers.

In summary, we report a role of the nutrient and stress-sensing TFEB in regulation of hepatic sulfur amino acid metabolism. By demonstrating that sulfur amino acid flux is tightly coupled to hepatic CoA production, our study provides new mechanistic insight underlying the importance of maintaining hepatic sulfur amino acid homeostasis in support of hepatic metabolic adaptation and redox homeostasis. Interestingly, both overnutrition and dietary protein deficiency appeared to be associated with disrupted hepatic sulfur amino acid homeostasis, rendering the liver a cysteine and CoA insufficient state that promotes maladaptation to increased lipid influx and oxidative stress.

## Methods
### Study approval
All animal protocols were approved by the Institutional Animal Care and Use Committee of the University of Oklahoma Health Sciences Center (#20-004) and the University of Kansas Medical Center (#2018-2457).

### Reagents
Anti-TFEB antibody (A303-673A, 1:1000 dilution) was purchased from Bethyl Laboratoryies, Inc. (Montgomery, TX). Actin antibody (ab3280, 1:10,000 dilution), and CBS antibody (ab135626, 1:2000 dilution), and CDO1 antibody (ab53436, 1:2000 dilution) was purchased from Abcam (Cambridge, MA). LC3 antibody (Cat #. 3868 S, 1:2000 dilution) was purchased from Cell Signaling Technology (Danvers, MA). P62 antibody (anti-SQSTM1, #89-015-843, 1:2000 dilution) was purchased from Fisher Scientific (Waltham, MA). MAT1A antibody (#712035, 1: 2000 dilution) and GNMT antibody (PA5-76962, 1: 2000 dilution) were purchased from ThermoFisher Scientific (Waltham, MA). Antibodies against GCLC (1:2000 dilution) and GCLM (1:2000 dilution) were provided by Terrrance J Kavanagh (University of Washington, Seattle, Washington, USA). Chloroquine was purchased from Sigma (St. Louis, MO). AST assay kit, total cholesterol assay kit and TG assay kit were purchased from Pointe Scientific (Canton. MI).

### Mice and treatments
WT male C57BL/6 J mice were purchased from the Jackson Lab (Bar Harbor, ME). Mice were housed in micro-isolator cages with Biofresh performance bedding (Pelleted cellulose) at -22 °C and -40–60% humidity under 7 a.m.–7 p.m. light cycle and 7 p.m.–7 a.m. dark cycle. Mice were fed one of the following diets ad libitum. The standard chow diet was PicoLab Rodent Diet 20 (#5053, LabDiet, St. Louis, MO)

containing 13 kcal% fat calories. Western diet (WD) contains 42 kcal% fat calories and 0.2% cholesterol (TD.88137, Envigo, Indianapolis, IN). Casein adjusted Western diets are custom made by Envigo and the dietary compositions are shown in Supplementary Table 2. Adenovirus and AAV vectors were injected via tail vein.

## VLDL-TG secretion and triglyceride and cholesterol analysis

To measure hepatic VLDL-TG secretion, mice were injected with 300 mg/kg tyloxapol via tail vein. Blood samples were collected immediately before and at 1 h and 2 h post tyloxapol injection for TG measurement. Liver tissues were used for lipid extraction in a mixture of chloroform: methanol (2:1; v- v). Lipid extracts were dried under nitrogen and resuspended in isopropanol containing 1% triton X-100. Total cholesterol and TG were measured with assay kits following the manufacturer's instruction.

## Metabolomics, statistical and bioinformatics analysis

For the bioinformatics analysis as shown in Fig. 1, the raw counts were normalized by first adjusting for instrument inter-day tuning differences by scaling the median count of each run-day block to 1 and then taking the log transformation. The changes in metabolite abundance among groups were modeled using two-way analysis of variance with statistical significance of pairwise comparisons calculated using orthogonal contrasts. The Benjamini-Hochberg method was used to adjust the resulting $p$ values to account for multiple hypothesis testing giving a false discovery rate (FDR, $q$ value)[57]. A fold-change greater than or equal to 1.5 and $q$ value less than or equal to 0.1 were considered significantly different. A global metabolite expression pattern was obtained by using factor analysis-based filtering as previously described[58]. A hierarchical clustering algorithm utilizing a Euclidian distance matrix (pairwise distance measure between metabolites across cohorts) and a Ward linkage function was used to cluster the metabolites. GraphPad Prism 6, R v3.5.1, and MATLAB R2017a software was used for the above analysis.

Metabolomics platform and analysis specified by Metabolon Inc. 1. Sample Preparation: Samples were prepared using the automated MicroLab STAR® system from Hamilton Company. Several recovery standards were added prior to the first step in the extraction process for QC purposes. To remove protein, dissociate small molecules bound to protein or trapped in the precipitated protein matrix, and to recover chemically diverse metabolites, proteins were precipitated with methanol under vigorous shaking for 2 min (Glen Mills GenoGrinder 2000) followed by centrifugation. The resulting extract was divided into five fractions: two for analysis by two separate reverse phase (RP)/ULPC–MS/MS methods with positive ion mode electrospray ionization (ESI), one for analysis by RP/UPLC-MS/MS with negative ion mode ESI, one for analysis by HILIC/UPLC–MS/MS with negative ion mode ESI, and one sample was reserved for backup. Samples were placed briefly on a TurboVap® (Zymark) to remove the organic solvent. The sample extracts were stored overnight under nitrogen before preparation for analysis. 2. Ultrahigh Performance Liquid Chromatography-Tandem Mass Spectroscopy (UPLC–MS/MS): All methods utilized a Waters ACQUITY ultra-performance liquid chromatography (UPLC) and a Thermo Scientific Q-Exactive high resolution/accurate mass spectrometer interfaced with a heated electrospray ionization (HESI-II) source and Orbitrap mass analyzer operated at 35,000 mass resolution. The sample extract was dried then reconstituted in solvents compatible to each of the four methods. Each reconstitution solvent contained a series of standards at fixed concentrations to ensure injection and chromatographic consistency. One aliquot was analyzed using acidic positive ion conditions, chromatographically optimized for more hydrophilic compounds. In this method, the extract was gradient eluted from a C18 column (Waters UPLC BEH C18-2.1 x 100 mm, 1.7 μm) using water and methanol,

containing 0.05% perfluoropentanoic acid (PFPA) and 0.1% formic acid (FA). Another aliquot was also analyzed using acidic positive ion conditions. However, it was chromatographically optimized for more hydrophobic compounds. In this method, the extract was gradient eluted from the same afore mentioned C18 column using methanol, acetonitrile, water, 0.05% PFPA and 0.01% FA and was operated at an overall higher organic content. Another aliquot was analyzed using basic negative ion optimized conditions using a separate dedicated C18 column. The basic extracts were gradient eluted from the column using methanol and water, however with 6.5 mM Ammonium Bicarbonate at pH 8. The fourth aliquot was analyzed via negative ionization following elution from a HILIC column (Waters UPLC BEH Amide 2.1 x 150 mm, 1.7 μm) using a gradient consisting of water and acetonitrile with 10 mM Ammonium Formate, pH 10.8. The MS analysis alternated between MS and data-dependent $MS^n$ scans using dynamic exclusion. The scan range varied slighted between methods but covered 70–1000 m/z. Raw data files are archived and extracted as described below. 3. Data Extraction and Compound Identification: Raw data was extracted, peak-identified and QC processed using Metabolon's hardware and software. These systems are built on a web-service platform utilizing Microsoft's.NET technologies, which run on high-performance application servers and fiber-channel storage arrays in clusters to provide active failover and load-balancing. Compounds were identified by comparison to library entries of purified standards or recurrent unknown entities. Metabolon maintains a library based on authenticated standards that contains the retention time/index (RI), mass to charge ratio (m/z), and chromatographic data (including MS/MS spectral data) on all molecules present in the library. Furthermore, biochemical identifications are based on three criteria: retention index within a narrow RI window of the proposed identification, accurate mass match to the library ± 10 ppm, and the MS/MS forward and reverse scores between the experimental data and authentic standards. The MS/MS scores are based on a comparison of the ions present in the experimental spectrum to the ions present in the library spectrum. While there may be similarities between these molecules based on one of these factors, the use of all three data points can be utilized to distinguish and differentiate biochemicals. More than 3300 commercially available purified standard compounds have been acquired and registered into LIMS for analysis on all platforms for determination of their analytical characteristics. 4. QA/QC: Several types of controls were analyzed in concert with the experimental samples: a pooled matrix sample generated by taking a small volume of each experimental sample (or alternatively, use of a pool of well-characterized human plasma) served as a technical replicate throughout the data set; extracted water samples served as process blanks; and a cocktail of QC standards that were carefully chosen not to interfere with the measurement of endogenous compounds were spiked into every analyzed sample, allowed instrument performance monitoring and aided chromatographic alignment. Supplementary Table 4 and Supplementary Table 5 describe these QC samples and standards. Instrument variability was determined by calculating the median relative standard deviation (RSD) for the standards that were added to each sample prior to injection into the mass spectrometers. Overall process variability was determined by calculating the median RSD for all endogenous metabolites (i.e., non-instrument standards) present in 100% of the pooled matrix samples.

## Targeted metabolite measurement by LC–MS

For sulfur amino acid and GSH analysis, samples were extracted in 90% acetonitrile and 10% water solution containing 0.1% formic acid and the internal standard L-Methionine-1-$^{13}$C. Following sonication and incubation on ice for 30 min, the sample extracts were centrifuged at 15,000 x g at 4 °C for 10 min. Supernatant was transferred to a new

tube and centrifuged at 15,000 x g at 4 °C for 10 min. Twenty μl of the clear supernatant was analyzed on a Thermo Scientific UltiMate 3000 UHPLC with a Waters XBridge Amide column (3.5 μm, 2.1 x 100 mm) and a TSQ Quantis triple quadrupole mass spectrometer coupled to a heated electrospray ion source (H-ESI) under the following conditions: Solvent A: 10 mM NH4FA (Ammonium Formate) in water (pH = 3.1); Solvent B: 10 mM Acetonitrile (pH = 3.1); Flow rate: 0.45 ml/min; Gradient Conditions (B%): 98% for 0.5 min, 98–40% in 5.5 min, hold 40% for 1 min, and then back to 98% in 0.2 min. Run time: 13 min. Column oven temperature: 40 °C. For CoA analysis, sample extracts were prepared in cold methanol containing the internal standard L-Methionine-1-$^{13}$C. After centrifugation, the supernatant was dried in a Speed-Vac and resuspended in a mixed solution of acetonitrile and water (2:1 ratio). A Thermo Scientific UltiMate 3000 UHPLC with a Waters Cortecs C18 column (2.7 μm, 4.6 x 100 mm) and a TSQ Quantis triple quadrupole mass spectrometer coupled to a H-ESI were used under the following conditions: Solvent A: 20 mM NH4Ac (Ammonium Acetate) in water; Solvent B: Acetonitrile; Flow rate: 0.3 ml/min; Gradient Conditions (B%): 0% for 1 min, 0–50% in 4 min, 50–95% in 1 min, hold for 1 min, 95–0% in 0.5 min. Run time: 16 min. Column oven temperature: 30 °C. Standard curves for each metabolite were generated with purified compounds and relative area under the curve (AUC) was calculated for detected metabolites in the samples.

## Chromatin immunoprecipitation assay

Pooled normal human livers (n = 3) (provided by the University of Kansas Liver Center) and chow-fed mouse livers (n = 3) were used to isolate nuclei as previously described[24]. The 3 human liver samples were from a 22 years old male (L949D), a 20 years old male (L394D) and a 42 years old female (L935D). These human liver samples were fully de-identified and were exempt from the University of Kansas Medical Center Institutional Review Boards approval. ChIP assays were performed with anti-TFEB antibody (A303-673A, 1:50 dilution, Bethyl Laboratoryies, Inc. Montgomery, TX), Dynabeads™ Protein G magnetic beads (ThermoFisher Scientific, Waltham, MA) and a ChIP assay kit (MilliporeSigma, Burlington, MA) following the manufacturer's instruction. Normal rabbit IgG (#2729, 1:50 dilution, Cell Signaling Technology, Danvers, MA) was used as negative control. Real-time PCR was performed to detect the relative abundance of the precipitated chromatin. ChIP real-time PCR primers are listed in Supplementary Table 2.

## Luciferase reporter constructs, cell culture and transient transfection

FLAG-tagged TFEB expression plasmid was a generous gift from Dr. Andrea Ballabio (Baylor College of Medicine, Houston, Texas). Human MAT1A promoter luciferase reporter constructs were described previously[59]. MAT1A promoter DNA fragments were synthesized (Supplementary Table 1) and inserted into PGL3-basic vector between the KpnI and HindIII sites. AML12 cells were a generous gift from Yanqiao Zhang (Northeast Ohio Medical University, Rootstown, OH). Further cell authentication was not performed. Cells were maintained in DMEM supplemented with 10% fetal bovine serum, 1% penicillin/streptomycin, and a mixture of insulin-transferrin-selenium (#41400-045, ThermoFisher, Waltham, MA). For transfection, cells are plated in 24-well plates. When cells were ~70% confluent, 0.2 μg/well luciferase reporter construct, 0.05 μg/well β-galactosidase expression construct and various amount of TFEB expression construct are co-transfected with Lipofectamine 3000 reagent following the manufacturer's instruction (ThermoFisher, Waltham, MA). PCDNA3.1 empty plasmid was added to normalize the total plasmid in each well. Cells are then cultured in serum free medium for 48 h. Luciferase activity and β-galactosidase activity are measured with Bright-Glo luciferase assay system and β-Galactosidase Enzyme Assay System, respectively (Promega, Madison, WI).

## Seahorse XF palmitate oxidation assay in intact cells

A Seahorse XFe96 Analyzer was used to measure oxygen consumption rate (OCR) using the Seahorse XF Palmitate Oxidation Stress Test Kit and FAO Substrate following the published protocol by the manufacturer (Agilent, Santa Clara, CA). Briefly, AML12 cells were seeded in XF96-well microplates at a density of 3 x 10$^5$ cells per well. After one day, cells were incubated overnight in substrate-limited growth medium (DMEM + 1 mM glutamine, 0.5 mM glucose and 0.5 mM XF L-carnitine, 1% FBS) with either 200 μM or 20 μM cystine and methionine. Cells were switched to fatty acid oxidation buffer one hour prior to assaying, consisting of 111 mM NaCl, 4.7 mM KCl, 1.25 mM CaCl$_2$, 2 mM MgSO$_4$, 1.2 mM NaH$_2$PO$_4$ supplemented with 2.5 mM glucose, 0.5 mM carnitine and 5 mM HEPES at pH 7.4. Oxygen consumption was measured in the presence of 166 μM palmitate conjugated to BSA (6:1 palmitate to BSA) or BSA alone. Following this, 1.5 μM oligomycin and 2 μM FCCP were sequentially added to uncouple mitochondria and maximal oxygen consumption was measured.

## Recombinant adenovirus and adeno-associated virus

First generation adenovirus vector in the DE1/E5 serotype 5 vector backbone was used to generate Ad-TFEB, Ad-shTFEB, Ad-scramble, and Ad-shPANK1. The same empty vector Ad-Null is used to inject control mice. Ad-Null and Ad-TFEB vectors contain a CMV promoter, while Ad-scramble, Ad-shTFEB and Ad-shPANK1 vectors contain U6 promoter. Adenovirus vectors were purchased from Vector Biolabs (Philadelphia, PA). Adenovirus was further amplified in HEK293A cells (Thermo Fisher Scientific, Grand Island, NY) and purified by using CsCl centrifugation followed by desalting with GE healthcare PD-10 Sephadex G-25 desalting columns (Thermo Fisher Scientific, Grand Island, NY). Adenovirus titer was determined with an Adeno-X rapid titer kit from Clontech (Mountain View, CA). Adeno-associated virus vectors AAV8-Null and AAV8-U6-shMAT1A were purchased from Vector Biolabs (Philadelphia, PA). AAV was used directly for injection.

## Immunoblotting

Liver tissues are homogenized in ice cold 1X RIPA buffer containing protease inhibitors followed by incubation on ice for 1 h. After centrifugation, supernatant is mixed with laemmli buffer, incubated at 95 °C for 5 min, and used for SDS-PAGE and immunoblotting. ImageJ2 software was used for densitometry analysis.

## Real-time PCR

Total liver RNA is purified with Trizol (Sigma-Aldrich, St. Louis, MO). Reverse transcription is performed with SuperScript III reverse transcriptase and Oligo dT primer following the manufacturer's instruction (Thermo Fisher Scientific, Grand Island, NY). Real-time PCR assay is performed with iQ SYBR Green Supermix (Bio-rad, Hercules, CA). Relative mRNA expression is calculated using the comparative CT (Ct) method and expressed as $2^{-\Delta\Delta Ct}$ with the control value arbitrarily set as 1. PCR primer sequences are listed in Supplementary Table 3.

## Statistics

Results were expressed as mean ± S.E.M or mean ± SD as specified in the Figure legends. One-way or two-way ANOVA and post hoc tests or Student's t-test (GraphPad Prism 6) was used to calculate the p value as noted in Figure Legends. A p < 0.05 was considered statistically significant. Bioinformatics and statistical analysis of the metabolomics data are described in Metabolomics and bioinformatics analysis in the Method section.

## Reporting summary

Further information on research design is available in the Nature Research Reporting Summary linked to this article.

## Data availability

The authors declare that all data supporting the findings of this study are available within the paper and its Supplementary Information files. Source data underlying all Figures are provided as a Source Data file. Source data are provided with this paper.

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

## Acknowledgements
This study is supported in part by NIH grant 1R01DK131064-01 (T.L.), 1R01 DK117965-01A1 (T.L.), 1R01 DK117418 (J.E.F.), U01 AA024733 (W.X.D.), R37 AA020518 (W.X.D.), R01 AG072895 (W.X.D.), and R01DK123763 (S.C.L.). We thank the Laboratory for Molecular Biology and Cytometry Research core facility, University of Oklahoma Health Sciences Center for developing the targeted LC–MS methods.

## Author contributions
D.M., H.W. Y.W., J.C., C.C., M.H., L.G. Y.D., D.C. and T.L. designed and performed the experiments. S.G. performed bioinformatics and statistical analysis of the metabolomics data. S.C.L. and W.X.D. provided key reagents. S.C.L., W.X.D. and J.E.F. reviewed the manuscript. T.L. supervised the study and wrote the manuscript.

## Competing interests
The authors declare no competing interests.
