## [Peer Review File · Nature Communications]

TFEB regulates sulfur amino acid and coenzyme A metabolism to support hepatic metabolic adaptation and redox homeostasisReviewer #1 (Remarks to the Author):

In this manuscript, the authors investigated the role of TFEB in regulation of hepatic sulfur amino acid and coenzyme metabolism, and how dysregulation of these metabolic pathways impact hepatic fatty acid oxidation and fatty liver formation. They first showed that overexpression of TFEB in the liver increases lysosomal proteolytic activity and induces hepatic sulfur amino acid metabolism. They then showed that TFEB transcriptionally promote MAT1A expression. They further showed that TFEB induces hepatic cysteine availability, enhances hepatic CoA production, and supports fatty acid oxidation. Moreover, they manipulated dietary protein availability and showed that dietary protein insufficiency selective reduces hepatic cysteine availability and impairs CoA synthesis and fatty acid oxidation. Finally, using long-term WD fed animals, they showed that diet-induced fatty liver is associated with insufficiency of cysteine and CoA.

General: This is an interesting study that includes several pieces of important data regarding the importance of hepatic sulfur metabolism in regulation of hepatic CoA production and fatty acid oxidation. Each piece of data is solid and convincing. However, several key links are missing to make this a cohesive story. The following concerns are needed to be addressed to strength the links between TFEB, sulfur amino acid metabolism, and CoA synthesis:

1. Figure 4, the described functional outcomes in RANK1 KO mice lack a clear link to TFEB-mediated hepatic CoA production. In order to put these two pathways together, the authors need to overexpress TFEB in control and RANK1 KO mice and show that TFEB overexpression fails to induce CoA enrichment in RANK1 KO mice. Of course, the best way to demonstrate the epistatic relationship between these two pathways is to inhibit an enzyme DOWNSTREAM of cysteine along the CoA synthesis pathway in control and TFEB overexpressing mice, and show that TFEB overexpression fails to induce CoA enrichment upon this enzyme inhibition.
2. Figure 5, again, the functional outcomes observed in MAT1A deficient mice lack a clear link to TFEB-mediated hepatic CoA production. The authors are encouraged to knock down MAT1A in TFEB overexpression mice to show that knockdown of this enzyme abolish TFEB overexpression-induced increase in CoA production and fatty acid oxidation.
3. Figure 6, what was the rationale to manipulate dietary cysteine availability using casein? It has been reported that casein is not a good dietary source of cysteine. Also, can dietary casein reduction blunt the impact of TFEB overexpression on CoA synthesis and fatty acid oxidation (repeat the diet feeding on control and TFEB overexpression mice)?
4. Figure 7, what is the link of this experiment to TFEB? What are the phenotypes of TFEB overexpression mice after 16 weeks of WD feeding?
5. Figure 8, again, as stated above, the epistatic relationship between TFEB and MAT1A, and between TFEB, cysteine availability, and fatty acid oxidation will need to be strengthened for the proposed model.

Minor:

1. Figure 3h and 3i, can the authors show the specificity of TFEB in the luciferase assay using a MAT1A promoter luc reporter lacking TFEB binding sites?
2. Figure 4c, the concentration unit of beta-hydroxybutyrate is missing on the y-axis.
3. Figure 5, what the dietary levels of cysteine in MAT1A knockdown experiment? Can restriction of dietary cysteine availability using defined diets exacerbate the observed metabolic defects?

Reviewer #2 (Remarks to the Author):

The present study aims at understanding the mechanism underlying liver steatosis induced by protein insufficiency. Starting from metabolomic analyses on livers of mice

injected with an adenoviral (Ad) vector expressing TFEB, the authors found that TFEB increases the hepatic content of sulfur amino acids that drives coenzyme A (CoA) production that in turns supports lipid degradation. They found that TFEB induces methionine adenosyltransferase (MAT1A) leading to increased cysteine/GSH pool that drives CoA synthesis. Consistently, they found that defects of hepatic CoA synthesis or reduced MAT1A exacerbate steatosis and liver injury. Mice fed with isocaloric protein-restricted diet were found to have hepatic cysteine deficiency, reduced CoA pool and acylcarnitine accumulation. These metabolic abnormalities were reversed by cystine supplementation.

The finding reported in this manuscript is potentially interesting. However, there are several missing experiments and controls that precludes the authors from drawing solid conclusions. For example, the authors argue that TFEB increased proteolysis and thus, several dipeptides are increased but none of the measured dipeptides includes methionine that should preferentially fluxed through the increased MAT1A upregulated by TFEB. Moreover, the study is lacking rescue of liver steatosis induced by MAT1A downregulation by the Ad-TFEB. This rescue should not be present in livers with PANK1 downregulation, as hypothesized by the authors based on the lack of effect of TFEB on PANK1 expression.

Does liver-specific knock-down of TFEB results in opposite changes in cysteine and coA? This study would support the conclusion about TFEB-mediated regulation of hepatic sulfur amino acid metabolism in preventing liver steatosis.

Do other inducers of liver autophagy result in the same metabolic derangements of cysteine and coA in livers? This is also important to understand whether the effect is specific for TFEB or common to other inducers (genetic and pharmacological) of liver autophagy, as suspected by the authors.

The authors are only measuring mRNA levels and they should confirm the effect of TFEB on both MAT1A and PANK by Western blots and enzyme activities.

Finally, a major limitation could be related to the adenoviral vector used (major issue#1) that might induce an inflammatory reaction that might be a confounding factor in the metabolomic analysis.

Major issues:

1. There are important information missing: what is the promoter driving the expression of TFEB in the Ad vector? What type of Ad vector (first generation Ad?) has been used? Although the authors used a control Ad-Null, if first generation adenoviral vector is used, the results would be complicated by viral gene expression that will eliminate transduced hepatocytes and induce an inflammatory reaction. The authors should clarify what they mean by Ad-null. All these important details are not present in the M&M.
2. Except for cysteine, most of the changes in fig. 2c are not significant or very mild. Why do authors not show Cys in the panel b of fig. 2? Several metabolites of Fig. 2a have not been measured. It would be helpful to show in fig. 2a which of the indicated metabolites were measured by metabolomic analysis and whether their concentrations were increased by TFEB. Why is taurine presented as a separate figure?
3. While the increase in Cys and homocysteine induced by TFEB is 2-3 fold, the increase in their product cystathionine is less than 1.5 fold. Moreover, SAH is not increase despite its substrate SAME is increased more than 4 fold. How can these unexpected concentrations be explained?
4. Upregulation of autophagy genes and cathepsin enzyme activities is expected and could be moved to supplementary.
5. Are the dipeptides shown in fig. 2C a representative subset of all dipeptides? What criteria were chosen to show that subset of dipeptides? Are longer (3-4 amino acids) peptides also increased? None of the dipeptides shown has cysteine. Does this mean that dipeptide with cysteine are not increased or were they not measured?
6. Does TFEB increase MAT1A protein amounts besides mRNA concentrations?
7. TFEB binding to the MAT1A promoter is poorly characterized. The putative TFEB binding sites on the MAT1A promoter should identified and mutagenized in the luciferase constructs to confirm transactivation. TFEB binding to the MAT1A promoter should also be confirmed by chromatin immunoprecipitation.

8. The knock-down experiment with Ad-shTFEB shows a trend in about 50% reduction of MAT1A mRNA. There is high variability that explain the not significant p-value but increasing the n might result in statistically significant difference.
9. CBS activity is hypothesized to be increased by allosteric activation of SAME. This hypothesis can be proven by measurement of CBS enzyme activity in liver extracts.
10. The findings of the experiments shown in fig 4D-H are expected since PANK downregulation is expected to result in impaired beta-oxidation and fat accumulation. The missing experiment is the delivery of TFEB in the livers of these mice to investigate whether the phenotype can be ameliorated.
11. Why AAV are used to knock-down MAT1A expression whereas PANK1 was downregulated by Ad vectors?
12. Oil red staining should be also presented besides H&E of livers of mice fed WD with downregulated MAT1A expression (as in fig. 4f).
13. In fig. 5m it is shown AST as marker of liver damage, but ALT would be a better marker.
14. Does the supplementation with Cys to the 5% protein diet prevent the liver steatosis? H&E and oil red staining should be performed.

Minor issues

1. Introduction: in citing previous studies investigating the role of TFEB in chronic liver diseases, the authors should also mention the clearance of mutant alpha1 antitrypsin globules and the improved ammonia detoxification by TFEB gene transfer (PMID: 23381957 and 29279371).
2. Introduction: why is dipeptide metabolism linked to sulfur amino acids? Please explain.
3. Acronyms should be spelled out the first time they appear in the text. This is not the case with 'WD'.

Reviewer #3 (Remarks to the Author):

In the manuscript authorized by Tiangang Li, the authors describe a possible new role for TFEB in the regulation of hepatic sulfur-amino acid metabolism. Although the results are interesting, the authors still need to further investigate how TFEB is able to regulate hepatic sulfur amino acid metabolism to show autophagy functionality in the different animal models used including MAT1A deficient, and those employed in fig 6 and 7, possible epigenetic changes due to changes in the SAME/ SAH and Hcy ratio, other pathways involved in GSH synthesis such as glutamine metabolism and pentose phosphate. TFEB levels, methionine cycle and restoration of liver physiology with cysteine supplementation in Fig. 6 and 7 should also be analyzed.

Finally, MAT1A deficient mice has been described as a model of VLDL impairment due to PEMT pathways dysfunction. This point should also be considered and analyzed in the MS. Is the NAFLD of MAT1A deficient mice under WD revert with cos supplementation

Specific questions and concerns are described below:

How is TFEB glutamine metabolism modulated in Figure 2? Are the levels of the enzymes regulated under the experimental conditions? Glutamine is directly involved in GSH synthesis.

Is TCA modulated under these conditions?

Given its importance in GSH synthesis, is pentose phosphate metabolism modulated under the control of TFEB?

What about the concentration of NADP+? Are ATP levels altered?

How are liver WD regulated in the presence or absence of TFEB, ALT and AST content, glucose and glycogen metabolism... what are the serum glucose levels in these mice? Mice with prolonged WD treatment in the presence or absence of TFEB should be performed to analyze the real effects of this transcription factor under experimental

conditions.

Based on the modulation of MAT1/III and SAME levels, GNMT levels, which are responsible for the conversion of SAME to SAH among other enzymes, and modulated sarcosine levels should be analyzed to obtain a comprehensive picture of methionine metabolism.

- In addition, given the high SAME levels, the amount of MTA and the activity of MTAP metabolism should be investigated, as well as DCSAM and polyamine metabolism.

Is global methylation regulated given the large amount of SAME and the small amount of SAH? Is there a possible link between cysteine metabolism and epigenetic changes?

In Fig. 3, is the protein abundance of MAT1/III modulated in the presence or absence of TFEB, not just the mRNA? Are GNMT protein levels regulated under these experimental conditions?

In Fig. 4, beta-oxidation activity and mitochondrial parameters such as mitochondrial ROS, ATP and OCR should be examined under these experimental conditions.

It has been previously reported that MAT1 deficiency leads to an alteration in PEMT metabolism that modulates VLDL export and results in TG accumulation PMID: 21837751 This aspect should be investigated in MAT1A-deficient mice. In addition, beta-oxidation and ketone bodies should be analyzed enzymatically in MAT1A-deficient mice. Is the NAFLD of MAT1A deficient mice under WD revert with cos supplementation

- Considering other mechanisms that may contribute to selective hepatic cysteine GSH in this context, glutamine metabolism and pentose phosphate should be analyzed (see Fig. 6).

-What are the levels of MAT1A and GNMT under the conditions described in Figs. 6 and 7? Are the levels of TFEB modulated? Are they also modulated in the context of altered diet? Is autophagy modulated under these conditions?

-Are there epigenetic changes in Fig. 7 when considering the SAM /SAH ratio? This hypothesis should be considered. Considering the results in Fig. 6 related to dietary manipulation, supplementation of Cys should be performed at 16 weeks WD to determine if lipid metabolism recovers.

Reviewer #4 (Remarks to the Author):

Minor comments on experimental aspects of the reviewed paper:

1. Information on caging conditions is not provided, such as light hours, feed availability (ad libitum?), etc. An accurate description of animal caging should be included in manuscript.
2. Please include the code/reference of the approved protocol by the Ethical Committee, as well as the affiliation of the Ethical Committee.
3. Instrumental conditions should be detailed, at least, in supplementary information. Chromatographic conditions regarding analytical column, mobile phase composition and gradient should be included, as well as injection volume and liver extraction procedure, highlighting extract composition and its compatibility with the used chromatographic run.
4. Critical information about HRMS instrument is missed and HRMS acquisition. Authors should describe which acquisition method is used (data-independent acquisition or data-dependent acquisition), as well as instrument resolution, mass range, and mass-axis calibration conditions.

5. It is important to describe, after indicate HRMS acquisition parameters, how metabolites were identified. In the manuscript is described that compounds were identified by the use of Metabolon's reference library, but it is not described which parameters were evaluated. I suppose that, at least, accurate-mass of (de)protonated metabolite and fragment ions were used in identification, but it would be interesting that authors include if chromatographic retention time was also evaluated. It is well-known that there are lots of isomeric metabolites that present similar fragmentation patterns and thus, retention time is pivotal for its identification.

6. As no standard curve was used, I assume that the identified metabolites presented the same instrumental behaviour than the information included in Metabolon's reference library. In some cases, chromatographic retention time can present shifts between the analytical standard and an authentic sample, being thus necessary to use spiked samples for an unequivocal metabolite identification. Can authors specify if all the 757 identified metabolites meet the analytical requirements to be considered as an unequivocal identification?

7. Moreover, for considering a metabolite as "identified", analytical reference standard of the compound should be run in the same analytical conditions than the samples. It should be indicated if all the 757 detected metabolites were identified by the use of analytical reference standards, or were only tentatively identified based on accurate-mass and fragment ions available on a reference library.

Manuscript # NCOMMS-21-40536B

We thank four reviewers for their constructive critics. During the revision, we have conducted additional in vitro and in vivo experiments and results have been incorporated in the revised manuscript. The revised manuscript contains 8 regular Figures, 11 Supplemental Figures and 3 Supplementary tables. Below we address each question raised by the reviewers in our point-to-point responses. All major changes are indicated in red font in the revised manuscript.

Reviewer #1

In this manuscript, the authors investigated the role of TFEB in regulation of hepatic sulfur amino acid and coenzyme metabolism, and how dysregulation of these metabolic pathways impact hepatic fatty acid oxidation and fatty liver formation. They first showed that overexpression of TFEB in the liver increases lysosomal proteolytic activity and induces hepatic sulfur amino acid metabolism. They then showed that TFEB transcriptionally promote MAT1A expression. They further showed that TFEB induces hepatic cysteine availability, enhances hepatic CoA production, and supports fatty acid oxidation. Moreover, they manipulated dietary protein availability and showed that dietary protein insufficiency selective reduces hepatic cysteine availability and impairs CoA synthesis and fatty acid oxidation. Finally, using long-term WD fed animals, they showed that diet-induced fatty liver is associated with insufficiency of cysteine and CoA.

General: This is an interesting study that includes several pieces of important data regarding the importance of hepatic sulfur metabolism in regulation of hepatic CoA production and fatty acid oxidation. Each piece of data is solid and convincing. However, several key links are missing to make this a cohesive story. The following concerns are needed to be addressed to strength the links between TFEB, sulfur amino acid metabolism, and CoA synthesis:

1. Figure 4, the described functional outcomes in PNAK1 KO mice lack a clear link to TFEB-mediated hepatic CoA production. In order to put these two pathways together, the authors need to overexpress TFEB in control and PANK1 KO mice and show that TFEB overexpression fails to induce CoA enrichment in PANK1 KO mice. Of course, the best way to demonstrate the epistatic relationship between these two pathways is to inhibit an enzyme downstream of cysteine along the CoA synthesis pathway in control and TFEB overexpressing mice, and show that TFEB overexpression fails to induce CoA enrichment upon this enzyme inhibition.

Response: The new data are presented in new Fig 4d-f in the revised manuscript. The data in original Fig 4d and 4e are moved to Supplemental Fig 6a-b.

We thank the reviewer's comment and would like to first clarify that the data presented in original Fig 4d-h are not aimed to suggest a TFEB - Pank1 axis that promotes CoA synthesis. Indeed, we have studied this possibility in several experimental settings and the results suggest that TFEB does not induce Pank1 expression (**Supplemental Fig 6c-d**). Instead, the main purpose of including this experiment is to demonstrate how modestly decreased CoA production due to liver Pank1 knockdown can increase hepatic steatosis susceptibility, therefore put an emphasis on the significance of hepatic CoA insufficiency in promoting steatosis. On the other hand, although Pank1 mediates the first step of CoA synthesis, its regulation at the transcription and enzyme activity level is still poorly defined. Importantly, the pathophysiological regulators of liver CoA synthesis are largely unknown. In this regard, our study suggests that cellular cysteine availability may drive CoA synthesis via a substrate-driven mechanism, which is a novel finding in the field. Now we have provided new data to show that TFEB stimulation of hepatic CoA synthesis is mediated via the synergistic action of two pathways: 1. Activation of autophagy-lysosome pathway to promote proteolysis to non-selectively produce amino acids including sulfur amino acids. And 2. Upregulation of MATA1 which stimulates the transsulfuration pathway to selectively enrich cellular cysteine pool. Please see our further responses below with new experimental data to support the role of these two pathways in mediating TFEB regulation of cellular CoA pool.

As suggested, we have conducted experiments in which we knocked down liver Pank1 and overexpressed TFEB in mice (**Fig 4d-f**). We found that knockdown of Pank1 increased liver TG accumulation in response to Western

diet challenge, but TFEB still reduced liver TG in Ad-shCon and Ad-shPank1 mouse livers. There are a few reasons that could explain why knockdown of liver Pank1 is not expected to abolish the anti-steatosis effect of TFEB. **1.** It should be noted that Pank1 is highly expressed in the liver, but there are mammalian Pank2 and Pank3 with the same function in CoA synthesis. Because of the expression of other isoforms of Pank, even complete deletion of Pank1 still leaves CoA synthesis active and therefore only decreases liver CoA pool by less than 50% (PMID: 20559429). Such reduction of CoA is sufficient to exacerbate liver steatosis under conditions of high CoA demand (as shown in prolonged fasting or WD challenge), but insufficient to abolish the beneficial effect of TFEB, not only because TFEB stimulation of CoA synthesis is PANK1 independent, but also because TFEB is known to improve liver metabolism and lower liver fat content by many other mechanisms including PGC1 α -PPAR α -mediated FAO, mitochondrial biogenesis, lysosome biogenesis, and autophagy, etc. (All these studies are cited and discussed in the text). These TFEB-activated catabolic pathways are expected to largely overcome partially decreased CoA synthesis by Pank1 knockdown in our model. **2.** We agree with the reviewer that blocking a CoA synthesis enzyme downstream of the cysteine addition step would be a better approach to test the TFEB-cysteine-CoA axis. However, to the best of our knowledge, no study has reported the outcome of complete absence of cellular CoA synthesis. This is likely because it is the general consensus that all cells can only acquire CoA via the highly conserved intracellular de novo synthesis. Therefore, completely blocking CoA synthesis (i.e. deletion of all Pank isoforms or downstream enzymes) is expected to abolish cellular CoA production but this may not be compatible with cell survival because CoA is required for a large number of critical cellular pathways in addition to FAO.

2. Figure 5, the functional outcomes observed in MAT1A deficient mice lack a clear link to TFEB-mediated hepatic CoA production. The authors are encouraged to knock down MAT1A in TFEB overexpression mice to show that knockdown of this enzyme abolish TFEB overexpression-induced increase in CoA production and fatty acid oxidation.

Response. The new data are added in Fig 5k-n and Supplemental Fig 7e-f in the revised manuscript.

We thank the reviewer's suggestion and have performed the experiment in which the TFEB effect on inducing cysteine, GSH and CoA production and lowering hepatic fat accumulation are compared in AAV-Null and AAV-shMAT1A treated mice fed WD for 2 weeks. Since several presented data suggest that diet is a significant source of cysteine and could mask the MAT1A-mediated effect, we fasted these mice for 16 h before analysis. Major findings include: 1. AAV-shRNA-mediated MAT1A knockdown prevented TFEB induction of MAT1A (Fig 5k). 2. Knockdown of MAT1A largely prevented TFEB induction of hepatic cysteine (Fig 5l). Given that a significant amount of cysteine is incorporated into GSH, we further found that MAT1A knockdown also abolished TFEB induction of γ -glutamylcysteine and GSH (Supplemental Fig 7 in the revised manuscript). 3. Consistent with cysteine as a key driver of CoA synthesis, TFEB induction of CoA was largely prevented upon MAT1A knockdown (Fig 5m). 4. Lastly, TFEB overexpression significantly reduced liver fat in AAV-Null groups. MAT1A knockdown significantly increased liver fat accumulation. However, TFEB still decreased liver TG in MAT1A knockdown livers although the liver TG did not decrease to the level seen in AAV-Null/Ad-TFEB group (Fig 5n), which is expected since TFEB is known to lower hepatic steatosis via other mechanisms. In summary, this set of new data suggests that during fasting MAT1A-driven methionine cycle/transsulfuration flux plays a significant role in mediating TFEB induction of liver cysteine and CoA. In addition, the close association of cellular cysteine and CoA again supports that cysteine is a major driver of cellular CoA synthesis. The lack of complete prevention of TFEB-mediated TG lowering effect in MAT1A deficient livers can be explained by other TFEB activated mechanisms such as PGC1 α -PPAR α axis, mitochondrial biogenesis and autophagy-lysosome pathways (references cited in text). It should also be noted that TFEB still

caused a modest increase of liver cysteine in MAT1A knockdown livers, suggesting that other mechanisms also mediates TFEB-induction of cellular cysteine. This is further addressed by studying the lysosome pathway in cultured cells treated with the lysosome inhibitor chloroquine (See new data in Supplemental Fig 8).

3. Figure 6, what was the rationale to manipulate dietary cysteine availability using casein? It has been reported that casein is not a good dietary source of cysteine. Also, can dietary casein reduction blunt the impact of TFEB overexpression on CoA synthesis and fatty acid oxidation?

Response: While there are other protein sources used in rodent diets, our lab has been using Western diet (Envigo TD.88137) with casein as the protein source to study fatty liver disease. Casein is a common protein source used in purified-ingredient research diets to allow for precise titration of amino acid concentration. Therefore, the protein adjusted diet is based on the Western diet formulation as shown in Supplemental Table 2. As we discussed in the text, it is technically challenging to specifically reduce dietary cysteine without altering other amino acid intake because casein is used in laboratory mouse diets as the sole source of protein. In contrast, it is possible to specifically supplement the mouse diet with cysteine in the form of cystine. We therefore designed the experiment whereby we first reduced overall dietary casein content which decreased all amino acid intake and then specifically restored cysteine intake without normalizing total protein intake in mice. In this model, we have found that lowering dietary protein intake markedly decreased hepatic cysteine pool, which correlated with impaired CoA synthesis and acylcarnitine accumulation. Interestingly, all these metabolic defects caused by low protein intake were rescued by dietary cystine supplement without normalizing total dietary protein intake. Therefore, the use of this dietary model has allowed us to obtain evidence supporting a key role of hepatic cysteine pool in maintaining hepatic CoA synthesis. As discussed in the revised manuscript, cells obtain cysteine via at least three major mechanisms: 1. Dietary uptake, 2. Synthesis from methionine via the transsulfuration pathway, and 3. Autophagy-lysosome proteolysis. In addition to dietary manipulation, we have also manipulated the transsulfuration pathway via MAT1A knockdown in vivo as shown above and the lysosome pathway via chloroquine treatment in cultured cells (Supplemental Fig 8). Findings from all three experimental conditions support the cysteine-CoA link.

We have performed the suggested experiment in which TFEB is over expressed in the liver of mice fed a 5% casein Western diet for 2 weeks, which we have shown to result in marked reduction in hepatic cysteine and GSH pool with minimal reduction of other amino acids. Interestingly, we found that the effect of TFEB to induce cysteine, GSH and CoA were significantly diminished in these mice (Figure for review, not included in revised manuscript). Furthermore, TFEB no longer reduced liver TG content under this condition. Although the underlying mechanisms are likely complex, we speculate that it is possible that the significant reduction of dietary casein protein (by 75%) and the subsequent depletion of hepatic cysteine and GSH pool by ~90% may be predominant and cannot be rescued by TFEB overexpression, especially given that both TFEB-stimulated pathways (Transsulfuration and lysosomal proteolysis) also rely on the availability of cellular protein and sulfur amino acids. However, the liver may alter overall metabolic status to adapt to significantly reduced dietary protein intake, and we can not rule out other underlying causes that remain to be further investigated. Due to the preliminary nature of these findings and the poorly understood mechanisms, we did not include these data in the revised manuscript but only presented here for review.

4. Figure 7, what is the link of this experiment to TFEB? What are the phenotypes of TFEB overexpression mice after 16 weeks of WD feeding?

Response. The purpose is to show that pathologically relevant NAFLD is a cysteine and CoA deficient condition. In terms of TFEB effect on the outcomes of long-term HFD feeding, it has already been reported that TFEB overexpression prevents NAFLD in chronic HFD-fed mice (Settembre C, et al, Nat Cell Biology, 2013, cited in the text). Consistently, we have also shown in Supplemental 1a-b that TFEB overexpression reduce liver fat in 6-week WD fed mice, suggesting that TFEB attenuates NAFLD development at the early stage of HFD feeding. Therefore, this study is mainly focused on the early metabolic changes to identify the TFEB-regulated mechanisms that provide metabolic benefits at later stage.

5. Figure 8, again, as stated above, the epistatic relationship between TFEB and MAT1A, and between TFEB, cysteine availability, and fatty acid oxidation will need to be strengthened for the proposed model.

Response. In the revised manuscript, we have provided several pieces of new in vitro and in vivo evidences to support an important role of the TFEB-MAT1A axis, autophagy-lysosome pathway, and dietary protein intake in regulating liver cysteine pool, CoA synthesis and hepatic steatosis. Taken together, these new data have strengthened the proposed model shown in Fig 8.

Minor:

1. Figure 3h and 3i, can the authors show the specificity of TFEB in the luciferase assay using a MAT1A promoter luc reporter lacking TFEB binding sites?

Response: TFEB has been shown to bind consensus E-box sequence that is common in the promoter of its target genes. In MAT1A, we have identified at least 6 putative TFEB binding sites in MAT1A promoter within the 1000 bp proximal promoter and it is possible that more TFEB bindings sites could be located further upstream in the 1000+ bp region. We reasoned that mutation of a single putative binding site could be compensated by the presence of other binding sites that help maintain MAT1A promoter activity. Therefore, we cloned each individual putative TFEB binding site in the luciferase reporter construct PGL3-basic and studied their role in mediating TFEB transactivation of the reporter activity. We found that two proximal binding sites in the human promoter (**Supplemental Fig 4c**) and one binding site in the mouse promoter (**Supplemental Fig 4d**) are functional but others have very weak to no effect in mediating TFEB transactivation of the reporter activity. In conclusion, these experiments have identified functional TFEB binding sites in the MAT1A proximal promoter. However, there could be other TFEB binding sites that also mediate TFEB induction of MAT1A transcription given that E-box sequences are common in many genes.

2. Figure 4c, the concentration unit of beta-hydroxybutyrate is missing on the y-axis.

Response: This result is part of the liver Metabolomics and only the relative levels can be calculated based on AUC with control value arbitrarily set as "1". This is clarified in the Figure legend.

3. Figure 5, what is the dietary levels of cysteine in MAT1A knockdown experiment? Can restriction of dietary cysteine availability using defined diets exacerbate the observed metabolic defects?

Response: In this set of experiments, the mice treated with AAV-Null or AAV-shMAT1A were fed a regular Western diet that contains 20% casein and 0.3% added methionine as protein source. As we discussed in the text, it is technically challenging to specifically decrease dietary cysteine intake without reducing overall dietary protein intake. We speculate that feeding liver MAT1A deficient mice with a protein deficient diet would exacerbate liver health since both conditions promote liver steatosis and injury. However, this would not specifically examine the role of cysteine in liver metabolism because both MAT1A deficiency and overall protein malnutrition are expected to affect hepatic metabolism via other cysteine-independent mechanisms. In addition, altered dietary protein intake is expected to affect extrahepatic tissues. Therefore, to address the effect of cysteine availability on liver lipid metabolism, we have performed in vitro cell culture experiments to show that specifically reducing exogenous cysteine supply decreases intracellular cysteine and CoA pool (**Supplemental Fig 8**) and causes lipid droplet accumulation as visualized by BODIPY staining of AML12 cells (**Fig 6k**), which supports our in vivo findings.

Reviewer #2

The present study aims at understanding the mechanism underlying liver steatosis induced by protein insufficiency. Starting from metabolomic analyses on livers of mice injected with an adenoviral (Ad) vector expressing TFEB, the authors found that TFEB increases the hepatic content of sulfur amino acids that drives coenzyme A (CoA) production that in turns supports lipid degradation. They found that TFEB induces methionine adenosyltransferase (MAT1A) leading to increased cysteine/GSH pool that drives CoA synthesis. Consistently, they found that defects of hepatic CoA synthesis or reduced MAT1A exacerbate steatosis and liver injury. Mice fed with isocaloric protein-restricted diet were found to have hepatic cysteine deficiency, reduced CoA pool and acylcarnitine accumulation. These metabolic abnormalities were reversed by cystine supplementation.

The finding reported in this manuscript is potentially interesting. However, there are several missing experiments and controls that precludes the authors from drawing solid conclusions.

1. For example, the authors argue that TFEB increased proteolysis and thus, several dipeptides are increased but none of the measured dipeptides includes methionine that should preferentially fluxed through the increased MAT1A upregulated by TFEB. Are the dipeptides shown in fig. 3C a representative subset of all dipeptides? What criteria were chosen to show that subset of dipeptides? Are longer (3-4 amino acids) peptides also increased? None of the dipeptides shown has cysteine. Does this mean that dipeptide with cysteine are not increased or were they not measured?

Response. Because there are 20 proteinogenic amino acids, there are theoretically 800 di-peptides that can be generated via proteolysis. However, our metabolomics only detected a total of 17 di-peptides (Shown in revised Fig2c and Supplemental Fig 3c), suggesting very limited detection due to LC/MS methods, sensitivity, and relative cellular abundance. Tri-peptides are not detected in this assay. Many other factors can affect the cellular abundance of di-peptides, including the peptide sequence of the degraded pool of proteins, the kinetics of dipeptidases that hydrolyze dipeptides to amino acids, and the pathways that consume specific di-peptides and amino acids, etc. Among the 17 dipeptides, our assay only detected one cysteine containing di-peptide cysteinylglycine which is increased by TFEB (Fig 2c). However, cysteinylglycine is also a GSH breakdown product and thus is not solely generated from protein breakdown.

Despite this limitation mainly due to the detection method, we want to clarify that this piece of data on di-peptide mainly serves as one of several pieces of evidences supporting overall increased lysosomal proteolysis by TFEB and it should be interpreted together with previous established role of TFEB in stimulating autophagy-lysosome pathway and other presented evidence in this paper that include:

- a. TFEB activation of autophagy and lysosomal biogenesis has been well documented in various cell types and we have shown that TFEB induced autophagy genes and cathepsin B activity in our model, suggesting activation of lysosomal function (Supplemental Fig 3a, 3b).
- b. Autophagy-lysosome pathway is a major mechanism mediating protein breakdown especially during fasting. Lysosomal proteolysis is known to generate a significant amount of di-peptides that is further converted to amino acids by dipeptidases (This is discussed in the text with supporting references cited).
- c. Activation of lysosomal proteolysis is a non-selective process, which means that methionine and cysteine have a similar chance as other amino acids to be released after a protein is degraded to subsequently

enter the cellular amino acid pool. In agreement, we have now provided new data to show that inhibition of lysosome function by chloroquine (CQ) significantly reduced cellular methionine and cysteine pool in cultured liver cells, especially under the culture condition where sulfur amino acid (SAA) methionine and cysteine are removed from the culture medium and the cells only rely on intracellular methionine and cysteine generation (**Supplemental Fig 8b, c, e**). These data serve as additional support that the lysosomal proteolysis contributes to intracellular methionine and cysteine pool. In addition, CQ treatment also reduced liver CoA, which supports the link between cysteine availability and CoA synthesis.

We also show that SAA deficiency promotes lipid droplet accumulation in liver cells (**Fig 6k**). However, the lysosomal proteolysis pathway does not explain selectively enriched cysteine. Instead, MAT1A induction by TFEB can further promote methionine conversion to cysteine via the methionine cycle/transsulfuration mechanism. In this revision, we have also shown in **Fig 5k-n** that knockdown of liver MAT1A can also prevent TFEB-mediated induction of cysteine and CoA in fasted mice (See our response to the next point). These new evidence support that TFEB could induce both the MAT1A pathway and the lysosome pathway to synergistically promote cellular cysteine and CoA production.

Supplementary Fig 8

2. Moreover, the study is lacking rescue of liver steatosis induced by MAT1A downregulation by the Ad-TFEB. This rescue should not be present in livers with PANK1 downregulation, as hypothesized by the authors based on the lack of effect of TFEB on PANK1 expression.

Response. The same questions are also asked by Reviewer #1. We have addressed these questions in response to the first reviewer's point #1 and point #2 with new data and detailed discussion, which are also presented below.

a. TFEB effect in liver MAT1A knockdown mice.

We thank the reviewer's suggestion and have performed the experiment in which the TFEB effect on inducing cysteine, GSH and CoA production and lowering hepatic fat accumulation are compared in AAV-Null and AAV-shMAT1A treated mice fed WD for 2 weeks. Since several presented data suggest that diet is a significant source of cysteine and could mask the MAT1A-mediated effect, we fasted these mice for 16 h before analysis. Major findings include: 1. AAV-shRNA-mediated MAT1A knockdown prevented TFEB induction of MAT1A (**Fig 5k**). 2. Knockdown of MAT1A largely prevented TFEB induction of hepatic cysteine (**Fig 5l**). Given that a significant amount of cysteine is incorporated into GSH, we further found that MAT1A knockdown also abolished TFEB induction of γ -glutamylcysteine and GSH (**Supplemental Fig 7 in the revised manuscript**). 3. Consistent with cysteine as a key driver of CoA synthesis, TFEB induction of CoA was largely prevented upon MAT1A knockdown (**Fig 5m**). 4. Lastly, TFEB overexpression significantly reduced liver fat in AAV-Null groups. MAT1A knockdown significantly increased liver fat accumulation. However, TFEB still decreased liver TG in MAT1A knockdown livers although the liver TG did not decrease to the level seen in AAV-Null/Ad-TFEB group (**Fig 5n**), which is expected since TFEB is known to lower hepatic steatosis via other mechanisms. In summary, this set of new data suggest that during fasting MAT1A-driven methionine cycle/transsulfuration flux plays a significant role in mediating TFEB induction of liver cysteine and CoA. In addition, the close association of cellular cysteine and CoA again supports that cysteine is a major driver of cellular CoA synthesis. The lack of complete prevention of TFEB-mediated TG lowering effect in MAT1A deficient livers can be explained by other TFEB activated mechanisms such

Figure 5

as PGC1 α -PPAR α axis, mitochondrial biogenesis and autophagy-lysosome pathways (references cited in text). It should also be noted that TFEB still caused a modest increase of liver cysteine in MAT1A knockdown livers, suggesting that other mechanisms also mediate TFEB-induction of cellular cysteine. This is further addressed by studying the lysosome pathway in cultured cells treated with the lysosome inhibitor chloroquine (See new data in Supplemental Fig 8).

b. TFEB effect in liver PANK1 knockdown mice.

As suggested, we have conducted experiments in which we knocked down liver Pank1 and overexpressed TFEB in mice (Fig 4d-f). We found that knockdown of Pank1 increased liver TG accumulation in response to Western diet challenge, and TFEB still reduced liver TG in Ad-shCon and Ad-shPank1 mouse livers as expected. Pank1 is highly expressed in the liver, but there are mammalian Pank2 and Pank3 with the same function in CoA synthesis. Because of the expression of other isoforms of Pank, even complete deletion of Pank1 still leaves CoA synthesis active and therefore only decreases liver CoA pool by less than 50% (PMID: 20559429). Such reduction of CoA is sufficient to exacerbate liver steatosis under conditions of high CoA demand (as shown in WD challenge), but insufficient to abolish the beneficial effect of TFEB not only because TFEB stimulation of CoA synthesis is PANK1 independent, but also because TFEB is known to improve liver metabolism and lower liver fat content by many other mechanisms including PGC1 α -PPAR α -mediated FAO, mitochondrial biogenesis, lysosome biogenesis, and autophagy, etc. (All these studies are cited and discussed in the text). These TFEB activated pathways are expected to largely overcome partially decreased CoA synthesis by Pank1 knockdown in our model.

3. **Does liver-specific knock-down of TFEB results in opposite changes in cysteine and CoA? This study would support the conclusion about TFEB-mediated regulation of hepatic sulfur amino acid metabolism in preventing liver steatosis.**

Response. We have conducted preliminary study in our liver TFEB knockdown mice and found that liver TFEB knockdown mice did not show reduced liver cysteine and only showed a modest ~12% reduction of liver CoA, which was not statistically significant (p=0.2). Similarly, the effect of TFEB knockdown on MAT1A expression was also modest (Fig 3c). As discussed in the revised text, we speculate that one possible explanation for such modest reduction of MAT1A may be that the TFEB knockdown is only partial and could be significantly compensated by other MiT family members including TFE3 and TFEC (supported by Pastore *et al.* PMID: 28283651, cited in revised text). In addition, MAT1A promoter is still poorly characterized and there could be other transcriptional factors, including those that recognize E-Box sites, that help maintain MAT1A expression when TFEB level is reduced. It is expected that cell may possess several compensatory mechanisms to help maintain intracellular sulfur amino acid pool due to their importance in generating many essential molecules including GSH, SAME, CoA, Taurine, etc.. Given the lack of changes at the metabolite level, further studies in this model were not pursued. In contrast, our studies in several other models including direct MAT1A knockdown in mouse livers, feeding mice a protein deficient diet, exposing cultured cells with cysteine deficient medium or chloroquine consistently support a cysteine-CoA link affected by MAT1A, autophagy and dietary cysteine availability. As discussed in Point #2a, our new data further shows that TFEB induction of cysteine and CoA is significantly attenuated in liver MAT1A knockdown mice (Fig 5k-n in revised manuscript), supporting the TFEB-MAT1A axis regulation of cysteine and CoA synthesis.

4. Do other inducers of liver autophagy result in the same metabolic derangements of cysteine and CoA in livers? This is also important to understand whether the effect is specific for TFEB or common to other inducers (genetic and pharmacological) of liver autophagy, as suspected by the authors.

Response: In this study we aim to address the role of two TFEB-stimulated pathways promoting cellular cysteine production to support CoA synthesis: 1. The transsulfuration pathways stimulated by MAT1A induction; and 2. Increased autophagy-lysosomal pathway that promotes proteolysis to generate methionine and cysteine. As pointed out by this reviewer, if the TFEB effect on cystine and CoA is partially mediated by its stimulation of autophagy-lysosome proteolysis, other autophagy-lysosome modulating agents should show similar effect on cellular cysteine and CoA. After careful consideration of a few available study models in our lab, we think that it may be challenging to address this question using in vivo genetic and pharmacological models of altered autophagy because of several reasons: 1. Currently, specifically targeting liver autophagy via pharmacological approach is still challenging because the effects of pharmacological activators of autophagy are usually not specific toward liver autophagy and can reduce liver steatosis via many other hepatic and extrahepatic mechanisms (recently reviewed in PMID: 35127371). In addition, we and others have shown that fatty liver is associated with autophagy impairment and cysteine and CoA deficiency (Fig 7 in revised manuscript). Therefore, pharmacological agents that reduce liver steatosis are expected to indirectly restore autophagy activity and cysteine and CoA levels, but this will only show correlation but does not establish causation. 2. Many other autophagy inducers do so by acting on cell signaling pathways. We found that inhibition of mTOR signaling in vitro strongly induces TFEB nuclear translocation and autophagy activation, but targeting mTOR signaling also affects many other cellular lipid and protein synthesis pathways, which makes result interpretation difficult. 3. We have limited access to genetic models with altered autophagy activity. We have shown that liver-specific ATG5 KO mice, which completely lack liver autophagy, also have persistent hepatic Nrf2 activation and GSH synthesis from cysteine (PMID: 22491424), which complicates the result interpretation.

To directly address this reviewer's question if autophagy-lysosome proteolysis is involved in maintaining cellular cysteine availability and if this is linked to cellular CoA synthesis, we treated cultured liver cells with lysosome inhibitor and measured its effect on cellular methionine, cysteine and CoA levels. This study in cultured liver cells allow us to avoid many confounding factors mentioned above and also other TFEB-mediated mechanisms that could affect cysteine and CoA. We have now shown (**Supplemental 8**) that upon acute inhibition of lysosome function (and therefore autophagy flux) by chloroquine (CQ) in cultured liver AML12 cells for a short period of 6 hours, cellular methionine and cysteine pool is significantly reduced. These changes are especially profound when sulfur amino acid (SAA) methionine and cystine are removed from the culture medium and the cells can only rely on intracellularly generated methionine and cystine. These data serve as additional support that the lysosomal proteolysis indeed contributes to intracellular methionine and cysteine pool. In addition, CQ treatment also reduced liver CoA, which is consistent with several other in vivo and in vitro data and support the link between cysteine availability and CoA synthesis. Culturing cells in SAA deficient medium also limits FAO capacity and promotes lipid droplet accumulation (**Fig 6k**). These data suggest that TFEB stimulation of autophagy-lysosome pathway likely contributes to cellular cysteine and CoA production. However, because lysosomal proteolysis is non-selective while TFEB induction of liver cysteine is more "selective" than its induction of other amino acids, we think that the lysosome activation and MAT1A induction by TFEB act in synergy to help maintain intracellular cysteine, GSH and CoA pool as shown in Fig 8.

Supplementary Fig 8

Figure 6

5. The authors are only measuring mRNA levels and they should confirm the effect of TFEB on both MAT1A and PANK by Western blots and enzyme activities.

Response. We have performed Western blots and confirmed that TFE B also induced MAT1A at the protein level. This data is now shown in **Supplemental Fig 4a**. We have also shown that TFEB causes significantly increased SAME (**Fig 2b**), suggesting that MAT1A enzyme activity is also increased as a result of increased proteon abundance. We have tried to measure PANK1 protein using several commercial antibodies. However, although a strong band that corresponds to the molecular weight of Pank1 was detected, we later used Pank1 KO mouse liver lysates (provided by Dr. Suzanne Jackowski, St Jude Children's Research Hospital) and found that they were non-specific bands that were also detectable in Pank1 KO mouse liver samples. Being a transcriptional factor, TFEB does not induce Pank1 mRNA (Supplemental Fig 6c-d in revised manuscript), indicating that TFEB does no directly regulate Pank1 via transcriptional activation.

6. Finally, a major limitation could be related to the adenoviral vector used that might induce an inflammatory reaction that might be a confounding factor in the metabolomic analysis. There are important information missing: what is the promoter driving the expression of TFEB in the Ad vector? What type of Ad vector (firs generation Ad?) has been used? Although the authors used a control Ad-Null, if first generation adenoviral vector is used, the results would be complicated by viral gene expression that will eliminate transduced hepatocytes and induce an inflammatory reaction. The authors should clarify what they mean by Ad-null. All these important details are not present in the M&M.

Response: We have provided the vector information in the revised manuscript. Both Ad-Null and Ad-TFEB are cloned in the DE1/E5 serotype 5 vector backbone (first generation), which is one of the most commonly used vector for gene delivery in mice. We fully agree with the reviewer and acknowledge that there is immune response after adenoviral vector-mediated liver gene delivery in mice. However, it is reported by many studies that the hepatic immune response to Ad-V is acute followed by a rapid decline typically within 2-3 days after delivery and significant immune responses are typically seen only when a high dose of adenovirus (10^{10} – 10^{11} pfu) is injected in mice. In contrast, we only used 5×10^8 pfu/mouse, which is far less than the Ad-V doses that induces significant liver inflammation and injury. One study (Crettaz et al, Hepatology, 2006, PMID: 16941711) showed that injecting mice adenovirus vector via i.v. caused liver injury at a dose of 10^{10} pfu but did not cause liver injury at a dose of 10^9 pfu. Another study using similar low dose as we used in mice showed that this dose causes minimal immune response in mouse liver (Matsuda et al, Lide Sci. 2021, PMID: 33412216). Consistent with these reports, we now show that Ad-Null and Ad-TFEB injected mice show similar ALT compared to mice without adenovirus injection (Supplemental Fig 1e). In further support, TFEB protein overexpression was verified in these studies (Supplemental Fig 1c in revised manuscript), which confirmed that transduced hepatocytes were not eliminated. Furthermore, we also show that the positive TFEB target genes in autophagy and lysosome biogenesis are all induced (Supplemental Fig 3), which is consistent with previous findings (Settembre et al, Science, 2011, PMID: 21617040) and demonstrates TFEB gain-of-function in mouse livers. Finally, we would like to acknowledge the limitation that the use of CMV promoter in Ad-TFEB could transduce non-parenchymal cells in the liver and our metabolomics analysis used whole liver tissue. However, it is well established that hepatocytes are highly active in sulfur amino acid, GSH and CoA metabolism and should be primarily responsible for the metabolite changes measured by metabolomics using whole liver tissues.

7. Except for cysteine, most of the changes in fig. 2c are not significant or very mild. Why do authors not show Cys in the panel b of fig. 2? Several metabolites of Fig. 2a have not been measured. It would be helpful to show in fig. 2a which of the indicated metabolites were measured by metabolomic

analysis and whether their concentrations were increased by TFEB. Why is taurine presented as a separate figure?

Response: We would like to clarify that all the metabolites shown in Figure 2 are from metabolomics analysis of the same set of liver samples and the data are presented in different panels. Cysteine is originally shown together with other amino acids to demonstrate that TFEB selectively induces liver cysteine but has a much milder effect in inducing other amino acids. We have now incorporated cysteine data in Fig 2b to plot all transsulfuration metabolites together. The missing metabolites in Fig 2a include DMG, betaine, CSA, and hypotaurine. CSA is not detected in the assay, while its downstream product hypotaurine and taurine are shown in Fig 2g. CDO1 protein is known to be stabilized by high cysteine (**Fig 2e**), which supports that cysteine flux through the taurine synthesis pathway is also increased by TFEB. The lack of free taurine increase is explained by their conjugation to bile acids in the text. Betaine and DMG are shown in **Supplemental Fig 2a**. Betaine and DMG levels are altered by TFEB and WD challenge but their ratio does not seem to change. Furthermore, because their levels can be affected by several other input and catabolism pathways, their changes cannot be solely explained by the homocysteine to methionine reaction. SAME is also linked to sarcosine and polyamine synthesis pathways, and the results are added in **Supplemental Fig 2b-c** in response to Reviewer #3's comment. These new data now provide a more comprehensive view of how TFEB alters the metabolites linked to hepatic sulfur amino acid metabolism. The result description is revised accordingly.

8. While the increase in Cys and homocysteine induced by TFEB is 2-3 fold, the increase in their product cystathionine is less than 1.5 fold. Moreover, SAH is not increase despite its substrate SAME is increased more than 4 fold. How can these unexpected concentrations be explained?

Response: We would like to speculate that the variations in fold induction of each intermediate metabolite in this pathway could be due to the differences in the kinetics of these reactions. For an example, SAH, once produced, may be rapidly converted to homocysteine. Given that our assays do not detect absolute concentration of each metabolite, the baseline concentration of each metabolite could also determine its fold induction by TFEB. Although we cannot rule out other causes since some of these sulfur amino acid metabolism intermediates are interconnected with many other cellular reactions, we think that the differences in metabolite abundance and kinetics can offer some possible explanations. Overall, since homocysteine and cystathionine are mainly derived from SAME, we think that the methionine cycle and transsulfuration are activated by TFEB despite the lack of marked elevation of SAH.

9. Upregulation of autophagy genes and cathepsin enzyme activities is expected and could be moved to supplementary.

Response: We agree and have moved these data to the Supplemental Fig 3.

10. TFEB binding to the MAT1A promoter is poorly characterized. The putative TFEB binding sites on the MAT1A promoter should identified and mutagenized in the luciferase constructs to confirm transactivation. TFEB binding to the MAT1A promoter should also be confirmed by chromatin immunoprecipitation.

Response: ChIP assays have confirmed TFEB enrichment on the MAT1A promoter chromatin in human and mouse livers (**Fig 3e, 3f**). We have now generated a series of MAT1A luciferase reporter constructs through which we have identified functional TFEB binding sites in the MAT1A gene promoter. These data are now shown in **Supplemental Fig 4c and 4d**. Please also see our response to Reviewer #1, minor point #1.

11. The knock-down experiment with Ad-shTFEB shows a trend in about 50% reduction of MAT1A mRNA. There is high variability that explain the not significant p-value but increasing the n might result in statistically significant difference.

Response: We thank the reviewer for the suggestion and have increased the sample size and repeated the measurement. The result now confirms that TFEB knockdown significantly reduced MAT1A mRNA expression (Fig 3c). The magnitude of reduction remains to be ~50% as previously observed. This modest reduction may be potentially explained by compensatory mechanisms discussed on our response to Point #3.

12. CBS activity is hypothesized to be increased by allosteric activation of SAME. This hypothesis can be proven by measurement of CBS enzyme activity in liver extracts.

Response: We show that TFEB does not affect liver CBS protein abundance (Supplemental Fig 4e). Our speculation that CBS activity may be activated by increased SAME is based on other previous in vitro studies using liver extracts or partially purified CBS. The CBS structural study also suggests SAME binding and allosteric activation of CBS (PMID: 25197074). However, we have measured CBS activity using liver extracts but failed to observe increased CBS activity by TFEB (Fig 3 for review only). Although this discrepancy could be due to the different assay systems used and the fact that previous in vitro assays were performed with added SAME while we measured CBS activity in liver extracts with increased endogenous SAME by TFEB. Given that our study focuses on TFEB induction of MAT1A to promote transsulfuration, we have deleted this statement in the “Result” section due to lack of supporting data.

13. The findings of the experiments shown in fig 4D-H are expected since PANK downregulation is expected to result in impaired beta-oxidation and fat accumulation. The missing experiment is the delivery of TFEB in the livers of these mice to investigate whether the phenotype can be ameliorated.

Response: We agree with this reviewer’s comment and have moved these data to Supplemental Fig 6a-b. We have added new data to show that PANK1 knockdown promote liver steatosis and TFEB can ameliorate PANK1 knockdown-induced liver fat accumulation (Fig 4d-f). This is further explained in our response to Point #2b and in the revised text.

14. Why AAV are used to knock-down MAT1A expression whereas PANK1 was downregulated by Ad vectors?

Response: Our lab has recently adopted the use of AAV vectors for liver gene delivery after the PANK1 knockdown experiments were already completed with Ad-shPANK1 a few years ago. Despite different vectors being used, the effect of liver Pank1 knockdown on liver steatosis is highly consistent with previously reported findings in Pank1 KO mice and chemical Pank1 inhibition studies that are cited in the text.

15. Oil red staining should be also presented besides H&E of livers of mice fed WD with downregulated MAT1A expression.

Response: We have presented both H&E and Oil Red O images in Fig 5g in the revised manuscript.

16. In fig. 5m it is shown AST as marker of liver damage, but ALT would be a better marker.

Response: We have measured ALT in these samples and found that ALT is also increased upon MAT1A knockdown (p=0.08). It is noted that both AST and ALT increased by less than 2 fold, which suggests very mild elevation likely because the mice mainly developed steatosis without marked injury at this time point. New data is now shown in Fig 5j in the revised manuscript.

17. Does the supplementation with Cys to the 5% protein diet prevent the liver steatosis? H&E and oil red staining should be performed.

Response: We have now shown in **Supplemental Fig 11** that livers from mice fed the 20% casein Western diet for 6 weeks have not developed profound hepatic steatosis due to the short duration of feeding. Interestingly, the livers from mice fed the 5% casein Western diet show pale hepatocyte staining throughout the liver. Dietary supplementation of cystine significantly prevented the low protein diet-induced liver histology change primarily around the centrilobular regions, but lipid droplets accumulation remains visible around the periportal areas. These findings suggest that cysteine deficiency partially contributes to low protein diet-induced steatosis, and overall protein deficiency also contributes to periportal steatosis that cannot be prevented by cystine supplementation. How overall dietary protein deficiency contribute to hepatic lipid accumulation via cysteine-independent mechanisms requires further investigation in the future. It should be noted that mice fed protein deficient diet are expected to have complexed metabolic changes in both hepatic and extrahepatic tissues. Given that we cannot specifically reduce methionine and cysteine in casein protein source in mouse diet, we further performed experiment in liver cells exposed to reduced methionine and cysteine without altering other amino acid concentration in culture medium. The results showed that this resulted in increased lipid accumulation (**Fig 6k**), which further supports our main conclusion.

Supplemental Figure 11

Figure 6

Minor issues

- 1. Introduction: in citing previous studies investigating the role of TFEB in chronic liver diseases, the authors should also mention the clearance of mutant alpha1 antitrypsin globules and the improved ammonia detoxification by TFEB gene transfer (PMID: 23381957 and 29279371).**

Response. These important new studies have been cited in the “Introduction” section.

- 2. Introduction: why is dipeptide metabolism linked to sulfur amino acids? Please explain.**

Response. As discussed earlier, only a small percentage of the cellular dipeptides are detected by metabolomics, but they are uniformly increased by TFEB. We think that increased dipeptides serve as a marker of increased lysosomal proteolysis, which is consistent with the established role of TFEB in stimulating autophagy and lysosomal biogenesis and the important role of the autophagy-lysosome pathway in generating intracellular dipeptides and amino acids. However, lysosomal proteolysis non-selectively generates intracellular amino acids, while activation of MAT1A and transsulfuration pathway can “selectively” enrich cellular cysteine pool. The two mechanisms may work in synergy. This is explained in the revised text as such: “*In addition to hepatic cysteine enrichment, we also observed that TFEB overexpression resulted in a general trend of increased amino acids despite being at a modest magnitude (Fig 2g). TFEB is a strong inducer of autophagy-lysosome pathway (16, 17). Consistently, we found that autophagy genes and lysosomal genes including proteases cathepsin B and cathepsin D were*

significantly induced upon TFEB overexpression (Supplemental Fig 3a). Further, liver cathepsin B activity, a marker for lysosomal proteolytic activity, was significantly elevated in TFEB overexpressing livers (Supplemental Fig 3b). Studies in intact livers showed that lysosomal proteolysis accounted for the majority of protein degradation to maintain the intracellular amino acid pool especially during fasting and nutrient deprivation (28-31). Studies with isolated liver lysosomes showed that lysosomal proteolysis generated a significant amount of dipeptides, which can be further hydrolyzed by dipeptidases to release amino acids (32-35). In addition to elevated amino acids (Fig 2g), we found that TFEB caused a general increase of most detected dipeptides by metabolomics in mouse livers (Supplemental Fig 3c). These findings provide additional evidence that TFEB induces hepatic lysosomal function to promote proteolysis. This mechanism may contribute to increased intracellular amino acids including methionine and cysteine but cannot fully account for the selectively enriched cysteine pool by TFEB (Fig 2b)."

- 3. Acronyms should be spelled out the first time they appear in the text. This is not the case with 'WD'.**

Response. This has been corrected.

Reviewer #3

In the manuscript authorized by Tiangang Li, the authors describe a possible new role for TFEB in the regulation of hepatic sulfur-amino acid metabolism. Although the results are interesting, the authors still need to further investigate how TFEB is able to regulate hepatic sulfur amino acid metabolism to show autophagy functionality in the different animal models used including MAT1A deficient, and those employed in fig 6 and 7, possible epigenetic changes due to changes in the SAME/ SAH and Hcy ratio, other pathways involved in GSH synthesis such as glutamine metabolism and pentose phosphate. TFEB levels, methionine cycle and restoration of liver physiology with cysteine supplementation in Fig. 6 and 7 should also be analyzed.

Finally, MAT1A deficient mice has been described as a model of VLDL impairment due to PEMT pathways dysfunction. This point should also be considered and analyzed in the MS. Is the NAFLD of MAT1A deficient mice under WD revert with cys supplementation.

Specific questions and concerns are described below:

1. How is glutamine metabolism modulated by TFEB in Figure 2? Are the levels of the enzymes regulated under the experimental conditions? Glutamine is directly involved in GSH synthesis. Is TCA modulated under these conditions? Given its importance in GSH synthesis, is pentose phosphate metabolism modulated under the control of TFEB? What about the concentration of NADP+? Are ATP levels altered?

Response: As shown in **Fig 2g** in revised manuscript, glutamine is modestly increased by TFEB as well as many other amino acids, which is possibly attributed to increased autophagy-lysosomal proteolysis. However, although glutamine can be converted to glutamate, glutamate is not induced by TFEB but tended to be slightly lower. It is possibly due to increased utilization for de novo GSH synthesis since γ -glutamylcysteine is markedly increased by TFEB (**Fig 2d**).

Glutamate can also be converted to α -ketoglutarate, through which glutamate feeds into TCA cycle. TCA intermediates are shown in Figure for review only. Citrate and aconitate, but not α -ketoglutarate, are induced by TFEB. It should be noted that our study measures steady state TCA intermediates and thus does not reflect TCA flux. However, previous studies have established a role of TFEB in inducing fatty acid oxidation and mitochondrial biogenesis (PMID: 23604321, PMID: 28011087), and we could reasonably speculate that TCA activity may be induced by TFEB. Metabolomics analysis does not detect ATP levels. We therefore measured ATP levels with an ATP assay kit and found that liver ATP levels

we therefore measured ATP levels with an ATP assay kit and found that liver ATP levels

were not altered by TFEB under chow or WD conditions (Figure for review only), suggesting that liver overall ATP is maintained.

In GSH synthesis, we show that GCLC and GCLM enzyme levels are not significantly induced by TFEB (Fig 2e in revised manuscript), suggesting that increased GSH synthesis is primarily driven by cysteine abundance. Limited metabolites in pentose phosphate pathway were detected by metabolomics and unfortunately the metabolomics methods were not able to detect NADP+ or NADPH. However, 6-phosphogluconate and ribose 1 phosphate, which is derived from ribose 5 phosphate, are not induced by TFEB under chow condition (Figure for review only), suggesting the pentose phosphate pathway, which generates NADPH to be used in GSSG-GSH conversion, may not be directly stimulated by TFEB. We also found that TFEB did not affect GSSG under chow condition and only reduced GSSG very modestly under WD condition (Fig 2c in revised manuscript). The slightly lower GSSG in WD condition could be due to reduced oxidative stress by TFEB. Given that GSSG concentration is usually 10% or less of GSH in the liver, we do not think that increased GSSG conversion to GSH could account significantly for the 3.5-fold increase of GSH by TFEB, especially considering the significantly increased cysteine and gamma-glutamyl-cysteine as markers of de novo GSH synthesis.

The new data on TCA cycle intermediates, liver ATP concentration and pentose phosphate pathway metabolites are mostly negative results and are therefore presented here for review but are not induced in the revised manuscript. However, we would be happy to include them if requested by this reviewer.

- How are liver WD regulated in the presence or absence of TFEB, ALT and AST content, glucose and glycogen metabolism... what are the serum glucose levels in these mice? Mice with prolonged WD treatment in the presence or absence of TFEB should be performed to analyze the real effects of this transcription factor under experimental conditions.

Response: Although most of our mechanistic studies are carried out in 1-2 weeks WD feeding models, we have also shown that TFEB reduced liver steatosis in 6 weeks WD-fed mice by ~60% in mice (Supplemental Fig 1b). TFEB does not affect blood glucose or ALT (Figures for review only), but it should be noted that at the early stage of WD feeding (6 weeks), the fasting blood glucose and transaminase levels are within normal range. On the other hand, hepatic TFEB activation has already been reported to prevent chronic HFD-induced NAFLD and hyperglycemia in mice (Nat Cell Biol. 2013 Jun; 15(6):647-58), which was later confirmed by other studies. Therefore, our study is mainly focused on the early effects of TFEB in order to identify the causative mechanisms by which TFEB activation prevents NAFLD at later stage. Given that TFEB does not alter fasting blood glucose, glycogen metabolism is not further investigated in these mice. However, studies have also shown that TFEB activation of the lysosomal pathway is beneficial in glycogen storage disease (PMID: 23606558), although this condition is not directly relevant to our study of NAFLD.

- Based on the modulation of MAT1/III and SAME levels, GNMT levels, which are responsible for the conversion of SAME to SAH among other enzymes, and modulated sarcosine levels should be analyzed to obtain a comprehensive picture of methionine metabolism.

Response: We have added these data in the revised manuscript. We found that GNMT is not altered by TFEB (**Supplemental Fig 4e**), suggesting that it is not a TFEB target gene. TFEB does not increase sarcosine levels in chow fed mice (**Supplemental Fig 2b**), which is consistent with the lack of TFEB induction of GNMT. We found that WD feeding decreased liver sarcosine levels, while the underlying mechanisms responsible for this changes remain to be determined. Relevant to such sarcosine change, we also show that DMG, which can be converted to sarcosine, is also lower in WD-fed mice (**Supplemental Fig 2a**). In WD fed mice, TFEB increased both sarcosine and DMG levels, which is consistent with the effect of TFEB in attenuating the WD-induced liver metabolic changes. Such discussion is added in the revised manuscript.

4. In addition, given the high SAME levels, the amount of MTA and the activity of MTAP metabolism should be investigated, as well as DCSAM and polyamine metabolism.

Response. New **Supplemental Fig 2c** has been added to show that MTA is induced 2 fold by TFEB in chow-fed mice but only 1.2 fold in WD mice. In polyamine synthesis, spermidine is not significantly altered but putrescine tended to be decreased in TFEB group, possibly due to increased conversion of putrescine to MTA. Although the metabolomics method failed to detect DCSAM, the observed changes of MTA and polyamines are consistent with increased SAME, which is in agreement with the knowledge that polyamine synthesis is a major source of MTA. These data also suggest that methionine salvage pathway may also be active in response to increased MTA.

5. Is global methylation regulated given the large amount of SAME and the small amount of SAH? Is there a possible link between cysteine metabolism and epigenetic changes?

Response: We agree with the reviewer that possible epigenetic effect of TFEB should be investigated due to increased SAME/SAH ratio by TFEB. To investigate this possible link, we have measured histone methylation mark H3K9 di/trimethylation and found that TFEB does not increase this histone methylation marker. It should be noted that, in addition to SAME/SAH ratio, global DNA and histone methylation is also regulated by other mechanisms (i.e. a large number of methylation and demethylation enzymes and their recruitment to specific transcriptional factors on the target gene chromatin), and these epigenetic changes are often differentially regulated among different subsets of genes. Therefore, it is not unexpected that increased SAME/SAH ratio does not correlated with globally increased histone methylation mark. We did not include this preliminary data in the revised manuscript and we think that TFEB regulation of global epigenetic changes is an interesting question that needs to be further investigated in more details in future studies.

6. In Fig. 3, is the protein abundance of MATI/III modulated in the presence or absence of TFEB, not just the mRNA? Are GNMT protein levels regulated under these experimental conditions?

Response: MAT1A protein is induced by TFEB but GNMT protein is not affected by TFEB in mouse livers. Please see new data in Supplemental Fig 4a, 4e.

7. In Fig. 4, beta-oxidation activity and mitochondrial parameters such as mitochondrial ROS, ATP and OCR should be examined under these experimental conditions.

Response: Our characterization data in liver PANK1 knockdown mice presented in the original manuscript were relatively limited because impaired mitochondrial fatty acid oxidation and increased liver steatosis in PANK1 deficient mice have been well established by a number of previous studies (PMID: 20559429, PMID: 17379144). The goals of using the liver PANK1 knockdown mice in our study are to first demonstrate the important effect of liver CoA insufficiency on hepatic steatosis development and to test if liver steatosis in PANK1 knockdown mice can be rescued by TFEB overexpression. These new data are presented in Fig 4d-f in the revised manuscript. Please also see our response to Reviewer #1 Point #1, which addresses the use of Pank1 knockdown mice in our study.

Here we have further addressed this reviewer's questions with additional experiments (**Figure for review**). We found that liver mitochondria from PANK1 knockdown mice showed reduced state 3 OCR. We measured liver 3-nitrotyrosine (3-NT) as an oxidative stress marker and found that PANK1 knockdown livers do not seem to show higher 3-NT levels. We also measured liver ATP levels and found that both groups show similar liver ATP concentration (123.8 ± 3.6 vs. 130.4 ± 12 nmole/g), suggesting that liver ATP is not reduced by PANK1 knockdown. To our surprise, plasma β -hydroxybutyrate (BHB), a marker for fatty acid oxidation, is not lower in PANK1 knockdown mice. However, fasting plasma glucose is significantly lower in PANK1 knockdown mice. Previous study reported that fasting hypoglycemia was a hallmark feature of PANK1 KO mice due to reduced hepatic fatty acid oxidation (PMID: 20559429). In general, these findings, together with increased liver steatosis in PANK1 knockdown livers presented in the revised manuscript Fig 4d-f, are consistent with previous findings in PANK1 KO mice. Therefore, these data are only presented here for review but are not included in the revised manuscript because previous studies of PANK1 deficient mice (PMID: 20559429, PMID: 17379144) have been cited in the text, and our data do not provide additional new information.

8. It has been previously reported that MAT1 deficiency leads to an alteration in PEMT metabolism that modulates VLDL export and results in TG accumulation (PMID: 21837751). This aspect should be investigated in MAT1A-deficient mice. In addition, beta-oxidation and ketone bodies should be analyzed enzymatically in MAT1A-deficient mice. Is the NAFLD of MAT1A deficient mice under WD revert with cys supplementation

Response:

- VLDL-TG secretion.** We are aware of this previous study (cited in the manuscript) and have further measured VLDL secretion in AAV-shMAT1A injected mice. We found that MAT1A deficient mice showed slightly but significantly increased VLDL-TG secretion (**Supplemental Fig 7d**), which is consistent with increased steatosis in this mice. In the previous study, MAT1A KO mice showed reduced VLDL-TG secretion at 3 months of age before they developed liver steatosis. However, MAT1A KO mice showed 2-fold increase in VLDL-TG secretion at 8 months of age after the MAT1A KO mice developed liver steatosis, suggesting that after steatosis developed, increased liver fat accumulation can drive VLDL-TG hyper-secretion in MAT1A KO mice. This is generally consistent with our MAT1A deficient mice challenged with WD that have worsened liver steatosis than controls. However, it should also be noted that the previous study used a complete MAT1A knockout mice since birth while we acutely but partially decreased liver

MAT1A in adult mice that were challenged with WD, which results in some difference in the course of steatosis development between the two models.

- Direct measurement of hepatic fatty acid oxidation in mice are still quite technically challenging in the field and we currently cannot perform this in vivo experiment in our lab. Alternatively, we have shown the β -hydroxybutyrate (BHB), a widely used surrogate marker for hepatic fatty acid oxidation, is reduced in MAT1A KD livers (**Fig 5f** in revised manuscript). In addition, we also measured plasma β -hydroxybutyrate and found that MAT1A KD mice show reduced plasma β -hydroxybutyrate (**Figure for review**).

- In our study, we used MAT1A deficient mice as a model to investigate its role in mediating TFEB regulation of CoA metabolism, but we have pointed out in the discussion that MAT1A deficiency alters lipid metabolism via many different mechanisms to contribute to the development of steatosis, liver injury and at later stage liver cancer. As several studies reported, some of these negative impact can be attenuated by SAME supplementation, but cystine supplementation will not increase SAME in MAT1A KO mice because the transsulfuration pathway is irreversible. In addition, long-term cystine supplementation in mice is expected to affect the metabolism of many extrahepatic tissues that could indirectly affect liver metabolism, preventing us to directly address the liver cysteine-CoA pathway. Instead of performing this in vivo experiment, we have now provided new in vitro data to show that directly reducing methionine and cysteine supply in cultured liver cells reduced cellular cysteine and CoA (**Supplemental Fig 8**) and promoted intracellular lipid droplet accumulation as revealed by BODIPY staining (**Fig 6k**). These data provide more straightforward evidence supporting a role of sulfur amino acids deficiency in promoting fat accumulation.

9. **Considering other mechanisms that may contribute to selective hepatic cysteine GSH in this context, glutamine metabolism and pentose phosphate should be analyzed (see Fig. 6).**

Response:

Glutamine is converted to glutamate that is a substrate for GSH synthesis. We found that 5% casein diet does not affect glutamine, glutamate or glycine levels while cysteine and gamma-glutamylcysteine were reduced (**Fig 6a, 6b, Supplemental Fig 9b**), suggesting that reduced cysteine, but not glutamine or glycine, limits de novo GSH synthesis. Unfortunately, this particular metabolomics assay failed to detect any key metabolites in the pentose phosphate pathway or NADP⁺ or NADPH that are involved in GSSG conversion to GSH. However, we found that both GSH and GSSG were markedly lower in 5% casein group (**Fig 6c, 6d**), suggesting the lower GSH is not due to higher GSH to GSSG conversion but is due to reduced de novo GSH synthesis.

10. What are the levels of MAT1A and GNMT under the conditions described in Figs. 6 and 7? Are the levels of TFEB modulated? Is autophagy modulated under these conditions?

Response:

- Fig 6. Protein adjusted WD fed mice. New data now show that MAT1A and GNMT levels were similar among all three groups (**Supplemental Fig 9e**). These data show that dietary protein deficiency does not affect the expression of these sulfur metabolism enzymes. We also measured nuclear and cytosolic TFEB and found no significant changes among the groups (**Supplemental Fig 9f**).

- Measurement of steady state autophagy marker LC3 in mouse livers does not accurately reflect autophagy flux. To determine how sulfur amino acid deficiency affects autophagy flux, we exposed liver cells with sulfur amino acid (SAA) methionine and cysteine deficient culture medium and measured autophagy flux under the treatment of chloroquine (CQ). Although it is well known that amino acid starvation induces autophagy, we found that sulfur amino acid deficiency per se did not induce autophagy flux because cells cultured in -SAA condition did not show increased p62 or LC3-II levels in the presence of CQ. This data is presented in **Supplemental Fig 8a**.

- In Fig 7, 16 week WD fed mice. The purpose of these data is to show that under NAFLD condition liver sulfur amino acid and CoA metabolism is disrupted. We now provide new data to show that MAT1A, GNMT and CBS are all downregulated to various degrees in NAFLD (Fig 7c in revised manuscript). We did not measure autophagy pathway because the liver autophagy impairment in NAFLD has been reported by many independent studies (recently reviewed in PMID: 34120768). Impaired hepatic TFEB in NAFLD has also been reported previously (PMID: 30044707). These points have been discussed in the revised manuscript as follows: “..... *reduced autophagic flux, lysosomal dysfunction, impaired TFEB activation, and dysregulation of sulfur amino acid metabolism have all been documented in experimental and human fatty livers (10, 25, 49, 58, 59).*”

11. Considering the results in Fig. 6 related to dietary manipulation, supplementation of Cys should be performed at 16 weeks WD to determine if lipid metabolism recovers.

Response:

- Reviewer #2 raised the same question and we performed additional experiments in 6 weeks WD-fed mice and the findings are as follows: Mice fed a 5% casein diet showed increased steatosis with uniformly pale hepatocytes and increased Oil Red O stain, consistent with protein deficiency-associated steatosis. Cystine supplementation largely prevented lipid droplet accumulation in the pericentral areas with remaining lipid droplet accumulation limited to the periportal areas (**Supplemental**

Fig 11). These results suggest that cystine supplementation can partially reverse dietary protein deficiency-induced liver steatosis, but dietary protein deficiency also promotes hepatic lipid accumulation around the periportal areas that are not reversed by cystine supplementation.

- The reason we did not perform 16 weeks WD study as suggested is because we have found that feeding mice a protein deficient diet for long term was detrimental to their health and growth, which cannot be rescued by cystine supplementation. Mice on 5% casein WD for long-term resemble malnutrition conditions with growth retardation that negatively impact various organs. This chronic model therefore is not suitable to address the liver sulfur amino acid metabolic pathways. We think that data obtained from short-term protein adjusted diet feeding with similar body weight among groups (**Supplemental Fig 9a**) best address the reviewer's question if cystine supplement reverses the protein deficiency effect on liver. Because dietary protein manipulation is expected to affect extrahepatic tissues, we have also performed cell culture studies to show that low sulfur amino acid (SAA) methionine and cysteine exposure promotes lipid droplet accumulation (**Fig 6k**), which serves as a "clean" system to demonstrate the link of cysteine deficiency to lipid accumulation.

12. Are there epigenetic changes in Fig. 7 when considering the SAM /SAH ratio? This hypothesis should be considered.

Response:

- The epigenetic changes, including DNA and histone methylation, in NAFLD is an important topic that has been investigated by many studies, which has been reviewed recently (PMID: 29122391; PMID: 27889327). As SAMe/SAH ratio clearly plays a role in epigenetic modifications, many other mechanisms, including a large number of epigenetic modification enzymes and their regulation by a host of transcriptional factors/signaling pathways, also play important roles in mediating differential epigenetic modifications affecting unique gene sets in NAFLD. As such, establishing a causative relationship between SAMe/SAH ratio and the epigenetic changes of a certain subset of genes is expected to be challenging and unlikely accomplished within the scope of this study. We agree with the reviewer that this is an important topic in the research field which can be further pursued in the future with the aid of global "omics" approaches and bioinformatics.

Reviewer #4

Minor comments on experimental aspects of the reviewed paper:

1. **Information on caging conditions is not provided, such as light hours, feed availability (ad libitum?), etc. An accurate description of animal caging should be included in manuscript.**

Response: Information on caging condition, dark/light cycle, and feed availability have been added.

2. **Please include the code/reference of the approved protocol by the Ethical Committee, as well as the affiliation of the Ethical Committee.**

Response: These information have been added to the Study approval as follow: "Study approval. All animal protocols were approved by the Institutional Animal Care and Use Committee of the University of Oklahoma Health Sciences Center (#20-004) and the University of Kansas Medical Center (#2018-2457)."

Regarding this reviewer's questions #3, #4, #5, #6 and #7 on metabolomics analysis methodologies, Metabolon Inc. has provided detailed information which addresses these questions. They are provided below and a Supplemental method file is included in the revised manuscript.

3. **Instrumental conditions should be detailed, at least, in supplementary information. Chromatographic conditions regarding analytical column, mobile phase composition and gradient should be included, as well as injection volume and liver extraction procedure, highlighting extract composition and its compatibility with the used chromatographic run.**
4. **Critical information about HRMS instrument is missed and HRMS acquisition. Authors should describe which acquisition method is used (data-independent acquisition or data-dependent acquisition), as well as instrument resolution, mass range, and mass-axis calibration conditions.**

Responses to point #3 and #4:

Sample Preparation: Samples were prepared using the automated MicroLab STAR® system from Hamilton Company. Several recovery standards were added prior to the first step in the extraction process for QC purposes. To remove protein, dissociate small molecules bound to protein or trapped in the precipitated protein matrix, and to recover chemically diverse metabolites, proteins were precipitated with methanol under vigorous shaking for 2 min (Glen Mills GenoGrinder 2000) followed by centrifugation. The resulting extract was divided into five fractions: two for analysis by two separate reverse phase (RP)/UPLC-MS/MS methods with positive ion mode electrospray ionization (ESI), one for analysis by RP/UPLC-MS/MS with negative ion mode ESI, one for analysis by HILIC/UPLC-MS/MS with negative ion mode ESI, and one sample was reserved for backup. Samples were placed briefly on a TurboVap® (Zymark) to remove the organic solvent. The sample extracts were stored overnight under nitrogen before preparation for analysis.

Ultrahigh Performance Liquid Chromatography-Tandem Mass Spectroscopy (UPLC-MS/MS): All methods utilized a Waters ACQUITY ultra-performance liquid chromatography (UPLC) and a Thermo Scientific Q-Exactive high resolution/accurate mass spectrometer interfaced with a heated electrospray ionization (HESI-II) source and Orbitrap mass analyzer operated at 35,000 mass resolution. The sample extract was dried then reconstituted in solvents compatible to each of the four methods. Each reconstitution solvent contained a series of standards at fixed concentrations to ensure injection and chromatographic consistency. One aliquot was analyzed using acidic positive ion conditions, chromatographically optimized for more hydrophilic compounds. In this method, the extract was gradient eluted from a C18 column (Waters UPLC BEH C18-2.1x100 mm, 1.7 µm) using water and methanol, containing 0.05% perfluoropentanoic acid (PFPA) and 0.1% formic acid (FA). Another aliquot was also analyzed using acidic positive ion conditions, however it was chromatographically optimized for more hydrophobic compounds. In this method, the extract was gradient eluted from the same afore mentioned C18 column using methanol, acetonitrile, water, 0.05% PFPA and 0.01% FA and was operated at an overall higher organic

content. Another aliquot was analyzed using basic negative ion optimized conditions using a separate dedicated C18 column. The basic extracts were gradient eluted from the column using methanol and water, however with 6.5 mM Ammonium Bicarbonate at pH 8. The fourth aliquot was analyzed via negative ionization following elution from a HILIC column (Waters UPLC BEH Amide 2.1x150 mm, 1.7 μ m) using a gradient consisting of water and acetonitrile with 10 mM Ammonium Formate, pH 10.8. The MS analysis alternated between MS and data-dependent MSⁿ scans using dynamic exclusion. The scan range varied slightly between methods but covered 70-1000 m/z. Raw data files are archived and extracted as described below.

5. **It is important to describe, after indicate HRMS acquisition parameters, how metabolites were identified. In the manuscript is described that compounds were identified by the use of Metabolon's reference library, but it is not described which parameters were evaluated. I suppose that, at least, accurate-mass of (de)protonated metabolite and fragment ions were used in identification, but it would be interesting that authors include if chromatographic retention time was also evaluated. It is well-known that there are lots of isomeric metabolites that present similar fragmentation patterns and thus, retention time is pivotal for its identification.**
6. **As no standard curve was used, I assume that the identified metabolites presented the same instrumental behaviour as the information included in Metabolon's reference library. In some cases, chromatographic retention time can present shifts between the analytical standard and an authentic sample, being thus necessary to use spiked samples for an unequivocal metabolite identification. Can authors specify if all the 757 identified metabolites meet the analytical requirements to be considered as an unequivocal identification?**

The following response address point #5 and #6:

Data Extraction and Compound Identification: Raw data was extracted, peak-identified and QC processed using Metabolon's hardware and software. These systems are built on a web-service platform utilizing Microsoft's .NET technologies, which run on high-performance application servers and fiber-channel storage arrays in clusters to provide active failover and load-balancing. Compounds were identified by comparison to library entries of purified standards or recurrent unknown entities. Metabolon maintains a library based on authenticated standards that contains the retention time/index (RI), mass to charge ratio (m/z), and chromatographic data (including MS/MS spectral data) on all molecules present in the library. Furthermore, biochemical identifications are based on three criteria: retention index within a narrow RI window of the proposed identification, accurate mass match to the library +/- 10 ppm, and the MS/MS forward and reverse scores between the experimental data and authentic standards. The MS/MS scores are based on a comparison of the ions present in the experimental spectrum to the ions present in the library spectrum. While there may be similarities between these molecules based on one of these factors, the use of all three data points can be utilized to distinguish and differentiate biochemicals. More than 3300 commercially available purified standard compounds have been acquired and registered into LIMS for analysis on all platforms for determination of their analytical characteristics.

We use QUICS (Quantify Individual Components in a Sample) methodology for rapid, in-depth evaluation of all possible metabolites.

References: PMID: 20955607 and PMID: 32445384

7. **Moreover, for considering a metabolite as "identified", analytical reference standard of the compound should be run in the same analytical conditions as the samples. It should be indicated if all the 757 detected metabolites were identified by the use of analytical reference standards, or were only tentatively identified based on accurate-mass and fragment ions available on a reference library.**

Response.

The vast majority of the metabolites are identified using Tier 1 identification. However, only a few metabolites without standards were identified based on accurate-mass and fragment ions available on a reference library. The below table explains how these metabolites are identified in the data table. Except

for general pathway analysis (presented in Fig 1) that include a few metabolites of none Tier 1 identification, all metabolites present in other data figures in the manuscript are Tier 1 identification.

Biochemical Name	Indicates a compound that has been confirmed based on an authentic chemical standard and we are highly confident in its identity. (Metabolomics Standards Initiative Tier 1 identification)
Biochemical Name*	Indicates a compound that has not been confirmed based on a standard, but we are confident in its identity. (Not Tier 1)
Biochemical Name**	Indicates a compound for which a standard is not available, but we are reasonably confident in its identity or the information provided. (Not Tier 1)

QA/QC: Several types of controls were analyzed in concert with the experimental samples: a pooled matrix sample generated by taking a small volume of each experimental sample (or alternatively, use of a pool of well-characterized human plasma) served as a technical replicate throughout the data set; extracted water samples served as process blanks; and a cocktail of QC standards that were carefully chosen not to interfere with the measurement of endogenous compounds were spiked into every analyzed sample, allowed instrument performance monitoring and aided chromatographic alignment. Tables 1 and 2 describe these QC samples and standards. Instrument variability was determined by calculating the median relative standard deviation (RSD) for the standards that were added to each sample prior to injection into the mass spectrometers. Overall process variability was determined by calculating the median RSD for all endogenous metabolites (i.e., non-instrument standards) present in 100% of the pooled matrix samples. Experimental samples were randomized across the platform run with QC samples spaced evenly among the injections, as outlined in Figure 1.

Table 1: Description of Metabolon QC Samples

Type	Description	Purpose
MTRX	Large pool of human plasma maintained by Metabolon that has been characterized extensively.	Assure that all aspects of the Metabolon process are operating within specifications.
CMTRX	Pool created by taking a small aliquot from every customer sample.	Assess the effect of a non-plasma matrix on the Metabolon process and distinguish biological variability from process variability.
PRCS	Aliquot of ultra-pure water	Process Blank used to assess the contribution to compound signals from the process.
SOLV	Aliquot of solvents used in extraction.	Solvent Blank used to segregate contamination sources in the extraction.

Table 2: Metabolon QC Standards

Type	Description	Purpose
RS	Recovery Standard	Assess variability and verify performance of extraction and instrumentation.
IS	Internal Standard	Assess variability and performance of instrument.

Reviewer #1 (Remarks to the Author):

The authors have made great efforts to address my major concerns, although not all of my concerns have been addressed. However, there are still a couple of claims that need to be clarified:

1. In the rebuttal letter, the authors stated that "we have also manipulated the transsulfuration pathway via MAT1A knockdown" (Page 3, on the response to the major point #4). In the main text (Line 193-197), despite the authors observed that "TFEB overexpression did not affect CBS and CSE mRNA or protein expression", they still conclude that "TFEB ... further stimulates methionine cycle and transsulfuration pathway to selectively enrich hepatic cysteine and GSH pool". CBS and CSE, but not MAT1A, are the direct players of this pathway (as the authors noted on Line 190-193). They therefore should clarify their statements in these two places: (1) MAT1A knockdown indirectly impacts the transsulfuration pathway; (2) TEFB does not directly regulate the transsulfuration pathway.

2. In the response to the minor point #3, the authors stated that "it is technically challenging to specifically decrease dietary cysteine intake without reducing overall dietary protein intake". However, Cysteine deficient defined diets are commercially available (e.g. Diet# 510027 from Dyets). So the authors need to rephrase this argument in the response as well as their discussion in the main text.

Reviewer #2 (Remarks to the Author):

The authors addressed several but not all the issues previously raised. The lack of increase in cysteine and coA in livers and the reduction on hepatic TG in mice with downregulation of MAT1A injected with Ad-TFEB is an important piece of data. However, the oil red staining of those mice should also be presented.

The issue of liver inflammation following intravenous injections of first-generation Ad vectors has not been satisfactorily addressed. The inflammatory response induced by first generation Ad vectors is biphasic: the first wave of inflammation occurring within hours after the injection is mediated by the viral capsid proteins and the second wave occurs weeks after the injection because of viral gene expression. The response provided by the authors about the low dose used (5×10^8 pfu/mouse) is puzzling because such low dose would not result in significant hepatic gene transfer.

The serum ALT shown in Supplemental Fig 1e is not informative unless the time post-Ad injection is shown.

Reviewer #3 (Remarks to the Author):

the authors answered each of the reviewer's questions

Reviewer #4 (Remarks to the Author):

The authors have addressed all the suggested comments. I am satisfied with the responses provided, and the paper can be now accepted.

Manuscript # NCOMMS-21-40536B

We thank all reviewers for their positive evaluation of our revised manuscript. We have addressed the additional comments below. The Oil Red O data has been added. The corresponding changes in the manuscript are labeled in red font.

Reviewer #1 (Remarks to the Author):

The authors have made great efforts to address my major concerns, although not all of my concerns have been addressed. However, there are still a couple of claims that need to be clarified:

1. In the rebuttal letter, the authors stated that "we have also manipulated the transsulfuration pathway via MAT1A knockdown" (Page 3, on the response to the major point #4). In the main text (Line 193-197), despite the authors observed that "TFEB overexpression did not affect CBS and CSE mRNA or protein expression", they still conclude that "TFEB ... further stimulates methionine cycle and transsulfuration pathway to selectively enrich hepatic cysteine and GSH pool". CBS and CSE, but not MAT1A, are the direct players of this pathway (as the authors noted on Line 190-193). They therefore should clarify their statements in these two places: (1) MAT1A knockdown indirectly impacts the transsulfuration pathway; (2) TEFB does not directly regulate the transsulfuration pathway.

Response. We agree with the reviewer and have revised the text, which now reads "*TFEB overexpression did not affect CBS and CSE mRNA or protein expression (Fig 3a, Supplementary Fig 4e), suggesting that TFEB may not directly induce transsulfuration pathway. Taken together, these results suggest a mechanism whereby TFEB induces hepatic autophagy and lysosome function to generate intracellular amino acids and further stimulates methionine cycle and synthesis of SAMe, which promotes downstream transsulfuration pathway to selectively enrich hepatic cysteine and GSH pool.*" We have also revised our previous response letter, which now reads "*we have also manipulated the transsulfuration pathway indirectly via MAT1A knockdown in vivo as shown above and the lysosome pathway via chloroquine treatment in cultured cells (Supplemental Fig 8).*"

2. In the response to the minor point #3, the authors stated that "it is technically challenging to specifically decrease dietary cysteine intake without reducing overall dietary protein intake". However, Cysteine deficient defined diets are commercially available (e.g. Diet# 510027 from Dyets). So the authors need to rephrase this argument in the response as well as their discussion in the main text.

Response. We thank the reviewer for pointing out the commercial source of the custom cysteine deficient diet, which we were not aware of previously. Accordingly, we have deleted this statement in our previous response letter. We did not make this statement in the revised main text, so there is no further change in the main text.

Reviewer #2 (Remarks to the Author):

1. The authors addressed several but not all the issues previously raised. The lack of increase in

cysteine and CoA in livers and the reduction on hepatic TG in mice with downregulation of MAT1A injected with Ad-TFEB is an important piece of data. However, the oil red staining of those mice should also be presented.

Response. We have performed the Oil Red O staining and the data is included in the new Supplementary Fig 7g. The Oil Red O staining results are consistent with our biochemical measurement of liver TG shown in Fig 5n.

2. The issue of liver inflammation following intravenous injections of first-generation Ad vectors has not been satisfactorily addressed. The inflammatory response induced by first generation Ad vectors is biphasic: the first wave of inflammation occurring within hours after the injection is mediated by the viral capsid proteins and the second wave occurs weeks after the injection because of viral gene expression. The response provided by the authors about the low dose used (5×10^8 pfu/mouse) is puzzling because such low dose would not result in significant hepatic gene transfer. The serum ALT shown in Supplemental Fig 1e is not informative unless the time post-Ad injection is shown.

Response. 1. We fully agree with the reviewer that Ad-V vectors are associated with early liver immune response and elevation of serum transaminases in mice. However, many detailed studies have shown that such response was highly dose-dependent and persistent hepatotoxicity was only seen when high dose of viral vector (i.e. 1×10^{10} PFU or higher) was used in mice (Crettaz et al, Hepatology, 2006, PMID: 16941711; Matsuda et al, Lide Sci. 2021, PMID: 33412216). When a low dose of Ad-V was used, an early response should still present but mild and transient. To minimize this liver immune response, we have titrated the dose of the Ad-TFEB to 5×10^8 PFU/mouse, which was significantly lower than the Ad-V dose (Ad-V at 10^{10} PFU or higher) that caused persistent hepatotoxicity in mice shown by other studies. We agree with the reviewer that a mild elevation of ALT is still expected immediately post injection of a low dose of Ad-V in our mice. However, as shown in Supplemental Fig 1e, the ALT measured at 2 weeks post injection of a low dose Ad-Null or Ad-TFEB was indistinguishable from that of the non-treated mice. These data suggest that the low dose Ad-V we used did not cause lasting hepatic immune response (second wave) in our model and the early response (first wave), although not immediately measured following the injection, may likely be very mild and transient. **2.** In terms of evidence supporting liver TFEB gain-of-function upon a low dose Ad-TFEB injection, we have shown that TFEB protein is increased by ~3 folds by Ad-TFEB injection at a dose of 5×10^8 PFU (Supplemental Fig 1c). In addition, we have shown that TFEB overexpression induced liver autophagy and lysosomal gene expression (Supplemental Fig 3a) and lysosomal cathepsin B enzyme activity (Supplemental Fig 3c), which are known TFEB regulated pathways. These data suggest that Ad-TFEB injected at a dose of 5×10^8 PFU is sufficient to result in liver TFEB gain-of-function in mice.

Reviewers 3 and 4 do not have additional comments.